# HEXST: Hexagonal Shifted-Window Transformer for Spatial Transcriptomics Gene Expression Prediction

Keunho Byeon [1]   Jin Tae Kwak [1]

## Abstract

Spatial transcriptomics offers spatially resolved gene expression profiling within tissue sections, but its cost and limited throughput hinder large-scale deployment. To extend this capability to routine practice, recent computational methods aim to infer spatial gene expression directly from ubiquitous hematoxylin and eosin-stained histology slides. However, most existing models assume Cartesian or geometry-agnostic locality, despite the hexagonal sampling of widely used spot-array platforms, and point-wise regression objectives often yield over-smoothed gene expression profiles, obscuring gene-specific spatial heterogeneity. To address these, we propose HEXST, a geometry-aligned Transformer for spatial gene expression prediction from histology. HEXST operates directly on hexagonal spot coordinates to enable efficient local-to-global contextual modeling via a tailored shifted-window attention mechanism and hexagonal rotary positional encoding. To enhance gene-wise spatial contrast, HEXST complements point-wise regression with a contrast-sensitive differential objective and transcriptomic priors from a pretrained single-cell foundation model during training. Across seven spatial transcriptomics datasets, HEXST consistently outperforms state-of-the-art models, providing accurate and robust spatial gene expression predictions while preserving gene-wise contrast and spatial heterogeneity.

## 1. Introduction

Spatial transcriptomics (ST) quantifies gene expression at spatially indexed locations within tissue sections, enabling the examination of cellular composition, tissue architecture, and disease microenvironments in situ (Ståhl et al., 2016; Marx, 2021; Moses & Pachter, 2022). Unlike bulk RNA-seq and dissociated single-cell RNA-seq, ST preserves spatial context and thus supports direct interrogation of biologically and clinically important phenomena such as tumor–stroma boundaries, immune infiltration patterns, and spatially organized niches. Rapid advances in both spot-based platforms (e.g., 10x Genomics Visium) and imaging-based assays (e.g., MERFISH) have further expanded the resolution and scale of spatially resolved transcriptomics (Rodriques et al., 2019; Chen et al., 2015; Marx, 2021).

Despite its impact, ST remains costly and relatively low-throughput, which limits its routine use in large clinical cohorts. In contrast, hematoxylin and eosin (H&E) stained histopathology is ubiquitous in clinical workflows, motivating a growing line of work that seeks to predict spatial gene expression directly from histopathology images (He et al., 2020; Pang et al., 2021; Xie et al., 2023), enabling spatially informed transcriptomic analyses even when only H&E slides are available. Existing approaches typically learn mappings from spot image to gene expression, and then incorporate broader context through attention-based mechanisms (e.g. Transformers) (Xiao et al., 2024), graph-structured spatial aggregation (Zeng et al., 2022), and neighbor-aware objectives (Qu et al., 2025).

While such methods have improved prediction performance, two fundamental challenges remain. First, the geometric sampling of widely used spot-array ST platforms (e.g., Visium) is non-Cartesian, capturing locations from approximately hexagonal lattices rather than a square grid. However, many existing methods impose neighborhood structure using priors that are either implicitly Cartesian (e.g., convolution operations, square attention windows, and Cartesian positional encodings) or geometry-agnostic (e.g., generic graph constructions). This mismatch alters what the model perceives as "local". Square-based partitioning can group non-neighboring spots while splitting immediate neighbors across different windows, yielding anisotropic and inconsistent receptive fields. Generic graphs can ignore hexagonal lattice structure and make connectivity sensitive to heuristic choices (e.g., neighborhood count and distance thresh-

---

[1]School of Electrical Engineering, Korea University, Seoul, Republic of Korea. Correspondence to: Jin Tae Kwak <jkwak@korea.ac.kr>.

*Proceedings of the 43rd International Conference on Machine Learning*, Seoul, South Korea. PMLR 306, 2026. Copyright 2026 by the author(s).

olds). As a result, geometry-consistent interactions must be learned largely from data rather than being encoded as an architectural inductive bias. Second, existing approaches often struggle to preserve gene-specific spatial heterogeneity, even when spot-wise metrics improve. In practice, training typically focuses on point-wise regression (e.g., per-spot error) with neighborhood-based aggregation. Since many genes are sparse and noisy, minimizing spot-wise error can favor predictions that regress toward local means, which can be exaggerated by neighborhood aggregation. Although locally averaged predictions may reduce point-wise error, they can also attenuate fine-scale variation and blur sharp boundaries, thereby weakening biologically meaningful signals that are inherently gene-specific and spatially heterogeneous. Important biological signals in spatial transcriptomics data lie not only in absolute expression values but also in spatial variations (Feng et al., 2024; Takano et al., 2024; Kueckelhaus et al., 2024). For example, tumor–stroma boundaries, immune niches, and other histological structures often manifest as sharp, abrupt changes in gene expression rather than gradual transitions. Accurate delineation of these spatial features is essential for identifying and analyzing localized cell-to-cell and microenvironmental interactions, and for discovering spatially restricted cellular states and biomarkers.

To address these limitations, we propose **HEXST**, a HEXagonal Shifted-window Transformer for spatial gene expression prediction. HEXST adapts the shifted-window Transformer to spot-array geometry by (1) constructing multiscale hexagonal attention windows that align with spot coordinates and (2) applying window partitioning and shifting across successive blocks, offering efficient local self-attention with cross-window information exchange and progressively expanding receptive fields from local to tissue-wide context. To encode relative spatial relationships under hexagonal coordinates, we introduce HexRoPE, a rotary positional embedding defined on hexagonal coordinates, which injects relative offsets into attention by rotating query/key features along the principal lattice directions. Finally, to better preserve spatial contrast and heterogeneity in gene expression, we propose a deviation-matching training objective that complements standard point-wise regression. It matches deviations between predictions and ground truth across spots, thereby discouraging collapse toward local averages and reducing over-smoothing.

## 2. Related Work

### 2.1. Spatial Gene Expression Prediction from Histology

A growing body of work aims to predict spatial gene expression from histopathology images to overcome the high cost and limited resolution of ST. Early studies primarily relied on convolutional neural networks (CNNs) or graph-based models to bridge image features and gene expression. ST-Net (He et al., 2020) introduces a CNN-based multi-branch architecture for spot-wise gene prediction from image patches, while TCGN (Xiao et al., 2024) formulates spatial relationships among spots as a graph to perform neighborhood-aware prediction. With the advent of Transformer architectures, Hist2ST (Zeng et al., 2022) employs an attention mechanism to capture long-range dependencies and introduces count-aware modeling of gene expression. HisToGene (Pang et al., 2021) leverages slide-level pretrained Vision Transformers to incorporate global tissue context into gene prediction. Exemplar-based approaches such as EGNv1 (Yang et al., 2023) and EGNv2 (Yang et al., 2024) further utilize retrieval-based transfer and graph-based spatial modeling to improve robustness. More recently, foundation models have been increasingly adopted for ST prediction. For instance, PEKA (Pan et al., 2025) uses parameter-efficient fine-tuning and knowledge distillation, in combination with PCA-Ridge regression for gene expression prediction. NH2ST (Qu et al., 2025) proposes a dual-branch framework with an image branch and a neighbor branch trained via contrastive learning.

### 2.2. Genomic Foundation Models

Genomic foundation models trained on large-scale single-cell and multi-omic datasets provide transferable transcriptomic representations for various downstream tasks, including cell type annotation, gene function prediction, and drug response prediction. Geneformer (Theodoris et al., 2023) employs a Transformer-based architecture to learn gene regulatory relationships, demonstrating strong performance in cell state and cell type classification. xTrimoscFoundation (Hao et al., 2024) is trained on diverse single-cell datasets to produce generalizable gene representations and has been successfully applied to spatial transcriptomics analysis and perturbation prediction.

### 2.3. Positional Encoding

Positional encoding is a critical component in Transformers for capturing spatial relationships. Vision Transformer (ViT) (Dosovitskiy, 2020) commonly utilizes Cartesian absolute positional encodings, which generalize poorly under layout changes. In contrast, rotary positional embedding (RoPE) (Su et al., 2024) and relative positional encoding (Shaw et al., 2018) have been proposed to encode relative offsets with attention in natural language processing. RoPE was originally designed for 1D sequence modeling and has been extended to 2D images and graph data (Heo et al., 2024). Nevertheless, these extensions generally assume Cartesian coordinates, limiting their ability to effectively represent the non-Cartesian spatial relationships induced by hexagonal spot-array ST platforms.

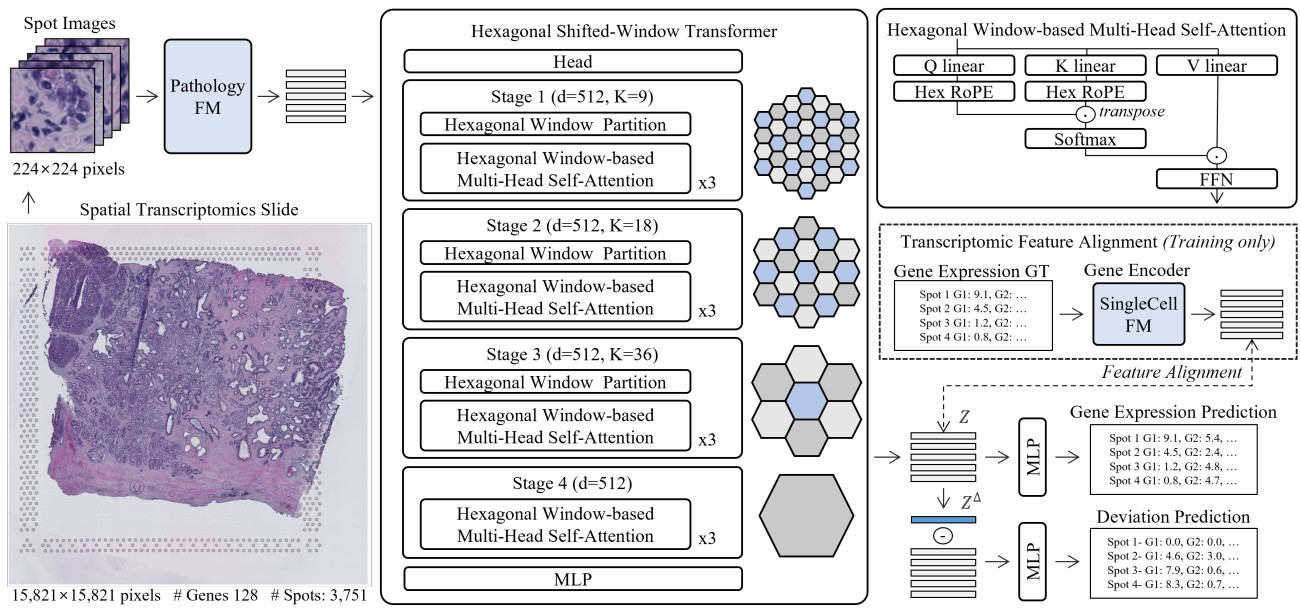

*Figure 1.* Overview of the HEXST architecture. Given a WSI, spot-level image patches are extracted and encoded by a pathology foundation model, and then processed by a multi-stage shifted hexagonal window Transformer with hexagonal rotary positional encoding (HexRoPE). HEXST predicts spatial gene expression from spot images by matching deviations between ground truth and predictions and aligning intermediate representations with transcriptomic embeddings during training.

## 3. Problem Formulation

Let an ST sample for a WSI comprises $N$ spatially indexed spots, where each spot $i \in \{1, \ldots, N\}$ is associated with a 2D Cartesian spatial coordinate $\mathbf{x}_i \in \mathbb{R}^2$, a gene expression vector $\mathbf{y}_i \in \mathbb{R}^G$ for $G$ genes, and a spot-centered histology image patch $\mathbf{I}_i$ cropped from the corresponding H&E WSI. An image encoder $E$ (e.g., a pretrained visual pathology foundation model) maps each patch $\mathbf{I}_i$ to a $D_{in}$-dimensional visual token $\mathbf{v}_i = E(\mathbf{I}_i) \in \mathbb{R}^{D_{in}}$. We collect the model input as $\mathbf{V} = \{\mathbf{v}_i\}_{i=1}^N$ and $\mathbf{X} = \{\mathbf{x}_i\}_{i=1}^N$ and denote the ground truth gene expression as $\mathbf{Y} = \{\mathbf{y}_i\}_{i=1}^N$.

Given paired training data $\{(\mathbf{V}_j, \mathbf{X}_j, \mathbf{Y}_j)\}_{j=1}^{N_w}$ for $N_w$ WSIs, the learning objective is to learn a gene expression predictor $f_\theta$ for all spots from spot-level visual tokens and their spatial coordinates:

$$\hat{\mathbf{Y}} = f_\theta(\mathbf{V}, \mathbf{X}), \qquad \hat{\mathbf{Y}} = \{\hat{\mathbf{y}}_i\}_{i=1}^N, \qquad \hat{\mathbf{y}}_i \in \mathbb{R}^G.$$

where $\hat{\mathbf{y}}_i$ is the predicted gene expression vector for spot $i$ and $\theta$ denotes the model parameters.

## 4. Method

In this section, we present HEXST, a geometry-aligned model for spatial gene expression prediction. HEXST comprises four key components: (1) a hexagonal shifted-window multi-head self-attention (HexMSA) that models non-Cartesian geometry of spot-array ST, (2) hexagonal

RoPE (HexRoPE) defined on axial/cube coordinates, (3) transcriptomic feature alignment that injects transcriptomic priors, and (4) deviation-matching mechanism that enhances spatial contrast. For each histology image of spot, a visual feature vector is extracted using the pathology foundation model UNI (Chen et al., 2024), and used as an input token to HEXST. The overall architecture is illustrated in Figure 1.

### 4.1. Hexagonal Shifted-Window Transformer

HEXST extends the shifted-window Transformer architecture to hexagonal spot-array transcriptomics (Figure 1). The model consists of four stages, each comprising three HexMSA blocks, and uses progressively larger hexagonal windows from Stage 1 to Stage 4 to expand the spatial receptive field. Within each stage, spots are partitioned into equal-sized hexagonal windows based on their coordinates, and HexMSA is computed independently within each window (i.e., over a hexagonal local neighborhood).

Let $\mathbf{H}^{(l,b)} \in \mathbb{R}^{N \times D}$ be the token features at stage $l \in \{1, \ldots, L\}$ and block $b \in \{1, \ldots, B\}$ (we use $L = 4$ and $B = 3$ blocks per stage). For block $b$ in stage $l$, we partition the $N$ spots into hexagonal windows with radius $K_l$ and shift parameter $\delta_b$:

$$\mathcal{W}^{(l,b)} = \Phi(\mathcal{X}, K_l, \boldsymbol{\delta}_b) = \{\Omega_m^{(l,b)}\}_{m=1}^M$$

where $\Phi(\cdot)$ denotes the spot partition function, $\Omega_m^{(l,b)} \subseteq \{1, \ldots, N\}$ is the index set of spots assigned to window $m$,

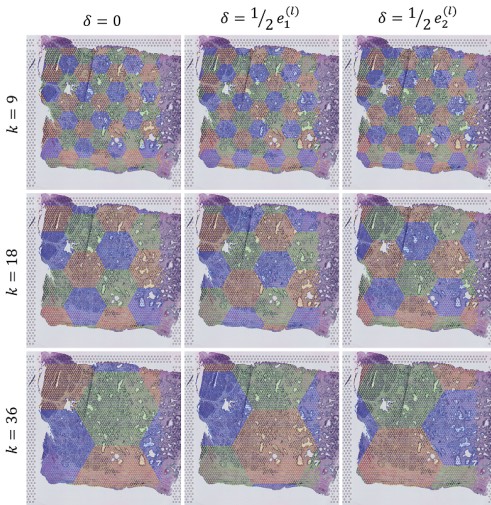

*Figure 2.* Visualization of hexagonal window partitioning. By varying the shifting offset $\boldsymbol{\delta}$ and window size $k$, distinct windows are formed (displayed with different colors).

and $M$ is the number of windows. The shifting parameter $\boldsymbol{\delta}_b$ offsets window centers across successive blocks, facilitating information exchange between neighboring windows and reducing boundary artifacts. Using $\Omega_m^{(l,b)}$, we assign the corresponding visual token features to the window $m$ and pack them into a fixed slot set $\mathcal{S}_{K_l}$, yielding $\mathbf{H}_m^{(l,b)} \in \mathbb{R}^{|\mathcal{S}_{K_l}| \times D}$, which is then processed by HexMSA:

$$\mathbf{H}_m^{(l,b)'} = \text{HexMSA}(\mathbf{H}_m^{(l,b)}) \in \mathbb{R}^{|\mathcal{S}_{K_l}| \times D}.$$

In the final stage, all spots are grouped into a single window to perform global self-attention and capture slide-level context. The resulting token representations are projected to output embeddings via a projection MLP: $\mathbf{Z} = \text{MLP}(\mathbf{H}^{(L,B)'}) \in \mathbb{R}^{N \times D_{out}}$, and spot-wise gene expression is predicted by a gene prediction head $\rho_{gene}$: $\hat{\mathbf{Y}} = \rho_{gene}(\mathbf{Z}) \in \mathbb{R}^{N \times G}$.

During training, we additionally compute centered token deviations $\mathbf{Z}^\Delta$ and predict gene expression deviations with a separate head $\rho_{dev}$: $\tilde{\mathcal{Y}}^\Delta = \rho_{dev}(\mathbf{Z}^\Delta)$, which supports the deviation-matching objective described in Section 4.2.3.

### 4.1.1. GEOMETRY ENCODING AND HEXAGONAL WINDOW PARTITIONING

In widely used ST platforms such as Visium, spot layouts approximately follow a hexagonal lattice structure with a near uniform 6-neighborhood structure. To align HEXST with this geometry, we map spot locations from Cartesian coordinates to a hexagonal lattice representation and construct multi-scale hexagonal windows.

Let $\{\mathbf{x_i}\}_{i=1}^N$ be the 2D Cartesian coordinates where $\mathbf{x_i} = $ $(x_i^1, x_i^2) \in \mathbb{R}^2$. We map each $\mathbf{x_i}$ to axial coordinates $(q_i, r_i)$ on a pointy-top hexagonal lattice. Since the observed inter-spot distances are not perfectly uniform, we estimate the slide length of the spot hexagonal lattice $s_{spot}$ by calculating the median neighbor distance and use it to normalize $\mathbf{x_i}$. The normalized coordinates are then converted to fractional axial coordinates and rounded to integer lattice coordinates by lifting to cube coordinates $(u_i, v_i, w_i) = (q_i, r_i, -q_i - r_i)$, which satisfies $u_i + v_i + w_i = 0$ and $(u_i, v_i, w_i)$ correspond to the principal directions of the hexagonal lattice.

At stage $l$, we construct hexagonal windows with radius $K_l$. We place window centers $\{\mathbf{c}_m^{(l)}\}_{m=1}^M$ on a coarser hexagonal lattice and assign each spot to its nearest center via Voronoi-style partition (Figure 2), yielding an equal-size hexagonal window partition $\mathcal{W}^{(l,b)} = \{\Omega_m^{(l,b)}\}_{m=1}^M$. For fixed-shape window self-attention, we represent each window by a shared set of discrete hexagonal slots indexed by integer axial offsets: $\mathcal{S}_{K_l} = \{(\Delta q, \Delta r) \in \mathbb{Z}^2 : \max(|\Delta q|, |\Delta r|, |\Delta q + \Delta r|) \leq K_l\}$. Each spot assigned to window $m$ is packed into $\mathcal{S}_{K_l}$ according to its local offset from the window center $\mathbf{c}_m^{(l)}$. Empty slots are masked and excluded from attention. Moreover, to facilitate cross-window information flow and mitigate boundary artifacts, we shift window centers across successive blocks:

$$\mathbf{c}_m^{(l)} + \boldsymbol{\delta}_m^{(l)}, \qquad \boldsymbol{\delta}_m^{(l)} \in \{\mathbf{0}, \frac{1}{2}\mathbf{e}_1^{(l)}, \frac{1}{2}\mathbf{e}_2^{(l)}\}$$

where $\mathbf{e}_1^{(l)} = (0, \sqrt{3}K_l \cdot s_{spot})$ and $\mathbf{e}_2^{(l)} = (\frac{3}{2}K_l \cdot s_{spot}, \frac{\sqrt{3}}{2}K_l \cdot s_{spot})$ are the basis vectors. Together, slot-based packing and shifted window partitions maintain computationally efficient window-based self-attention while encouraging cross-window interaction. Details of geometry encoding and hexagonal window partitioning are provided in Appendix A.

### 4.1.2. HEXAGONAL WINDOW-BASED MULTI-HEAD SELF-ATTENTION (HEXMSA)

HEXST performs HexMSA independently within each hexagonal window using a fixed slot layout $\mathcal{S}_{K_l}$, supporting efficient batched computation. Given $\mathbf{H}_m^{(l,b)} \in \mathbb{R}^{|\mathcal{S}_{K_l}| \times D}$, multi-head projections with $N_h$ heads produce queries $\mathbf{Q}_h$, keys $\mathbf{K}_h$, and values $\mathbf{V}_h$, where $\mathbf{Q}_h, \mathbf{K}_h, \mathbf{V}_h \in \mathbb{R}^{|\mathcal{S}_{K_l}| \times D_H}$, $h = 1, \ldots, N_h$, and $D_H$ denotes the per-head dimension. To encode relative hexagonal geometry, we apply HexRoPE (defined in Section 4.1.3) to the query and key features for each head:

$$\tilde{\mathbf{Q}}_h = \text{HexRoPE}(\mathbf{Q}_h), \qquad \tilde{\mathbf{K}}_h = \text{HexRoPE}(\mathbf{K}_h).$$

Window self-attention is then computed as $\text{softmax}(\tilde{\mathbf{Q}}_h \tilde{\mathbf{K}}_h^\top / \sqrt{D_H})\mathbf{V}_h$, followed by a feed-forward network with residual connections and layer normalization. The outputs from all heads are concatenated.

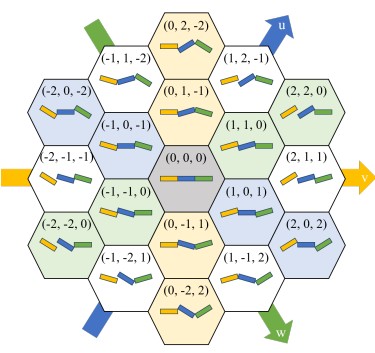

*Figure 3.* Conceptual illustration of hexagonal rotary positional encoding (HexRoPE). Within each window, HexRoPE assigns a unique set of relative spatial offsets that are used to rotate query and key features. Rectangular boxes denote rotation angles applied along each principal direction $(u, v, w)$ of the hexagonal lattice.

### 4.1.3. HEXAGONAL ROTARY POSITIONAL ENCODING (HEXROPE)

Absolute (Cartesian) positional encoding used in standard Transformers do not naturally respect hexagonal geometry. Relative positional encoding such as RoPE (Su et al., 2024) has shown to be effective in large language modeling. To integrate this relative scheme to hexagonal lattice, we propose HexRoPE that injects relative positional information directly into attention by rotating query and key features according to relative spatial offsets, preserving translation equivariance within each window.

Let $\mathbf{h} \in \mathbb{R}^D$ be a per-head token feature (used for query and key projections), which is associated with cube offsets $(\Delta u, \Delta v, \Delta w) = (\Delta q, \Delta r, -(\Delta q + \Delta r))$. We distribute its channels across three cube axes as evenly as possible: $\mathbf{h} = \left[\mathbf{h}^{(u)}; \mathbf{h}^{(v)}; \mathbf{h}^{(w)}; \mathbf{h}^{(\mathrm{rem})}\right]$ where $\mathbf{h}^{(\mathrm{rem})}$ contains any remaining channels as $D$ is not divisible by three. For each axis $\alpha \in \{u, v, w\}$, we follow RoPE (Su et al., 2024) to rotate the corresponding features $\mathbf{h}^{(\alpha)}$ using the corresponding cube-coordinate offset $\Delta \alpha$:

$$\begin{bmatrix} \tilde{\mathbf{h}}_{2k}^{(\alpha)} \\ \tilde{\mathbf{h}}_{2k+1}^{(\alpha)} \end{bmatrix} = \begin{bmatrix} \cos\theta_k^{(\alpha)} & -\sin\theta_k^{(\alpha)} \\ \sin\theta_k^{(\alpha)} & \cos\theta_k^{(\alpha)} \end{bmatrix} \begin{bmatrix} \mathbf{h}_{2k}^{(\alpha)} \\ \mathbf{h}_{2k+1}^{(\alpha)} \end{bmatrix},$$

with frequencies:

$$\omega_k = \mathtt{base}^{-2k/D_c}, \quad \theta_k^{(\alpha)} = \Delta\alpha\, \omega_k, \quad k = 0, \ldots, \frac{D_c}{2} - 1,$$

where $\mathtt{base} = 10000$ and $D_c$ denotes the number of channels assigned to each axis. The HexRoPE-rotated feature is obtained by concatenation: $\tilde{\mathbf{h}} = \left[\tilde{\mathbf{h}}^{(u)}; \tilde{\mathbf{h}}^{(v)}; \tilde{\mathbf{h}}^{(w)}; \mathbf{h}^{(\mathrm{rem})}\right]$.

### 4.2. Training Objective

We train HEXST to predict spot-wise gene expression while maintaining gene-specific spatial contrast.

#### 4.2.1. SPOT-WISE AND GENE-WISE LOSS

We adopt mean squared error (MSE) to penalize spot-wise prediction errors:

$$\mathcal{L}_{\mathrm{MSE}} = \frac{1}{NG} \sum_{i=1}^{N} \|\hat{\mathbf{y}}_i - \mathbf{y}_i\|_2^2 .$$

To encourage agreement of gene-wise spatial expressions across spots, we optimize a Pearson correlation loss:

$$\mathcal{L}_{\mathrm{PL}} = 1 - \frac{1}{G} \sum_{g=1}^{G} \mathrm{PCC}(\hat{\mathbf{y}}_{\cdot,g}, \mathbf{y}_{\cdot,g}),$$

where $\hat{\mathbf{y}}_{\cdot,g}$ and $\mathbf{y}_{\cdot,g}$ denote the predicted and ground-truth expression vectors across all spots for gene $g$, respectively.

#### 4.2.2. TRANSCRIPTOMIC FEATURE ALIGNMENT LOSS

To inject transcriptomic priors, we regularize HEXST by aligning its output embeddings with transcriptomic embeddings obtained from a pretrained single-cell foundation model (scFoundation) (Hao et al., 2024).

For each spot $i$, let $\mathbf{z}_i \in \mathbb{R}^{D_o}$ denote the output embedding of HEXST, and let $\mathbf{t}_i \in \mathbb{R}^{D_t}$ denote the corresponding transcriptomic embedding from scFoundation. $\mathbf{t}_i$ is obtained by feeding the ground truth gene expression values into scFoundation, where genes absent from the model's vocabulary are set to zero. Since $\mathbf{z}_i$ and $\mathbf{t}_i$ may differ in dimensionality and scale, we apply a learnable linear projection $p(\cdot): \mathbb{R}^{D_o} \to \mathbb{R}^{D_t}$ and optimize a cosine alignment loss:

$$\mathcal{L}_{\mathrm{TFA}} = \frac{1}{N} \sum_{i=1}^{N} \left(1 - \frac{p(\mathbf{z}_i)^\top \mathbf{t}_i}{\|p(\mathbf{z}_i)\|_2 \|\mathbf{t}_i\|_2}\right).$$

We note that transcriptomic embeddings are used only during training.

#### 4.2.3. DEVIATION-MATCHING LOSS

Standard point-wise regression losses (e.g., MSE) encourage accurate reconstruction of absolute gene expression values but can inadvertently favor spatially over-smoothed predictions, obscuring gene-specific contrast and heterogeneity. To explicitly preserve relative spatial variation, we introduce a deviation-matching loss that aligns per-gene relative variation patterns across spots. For each gene $g$, we compute the ground-truth gene-wise deviations by subtracting the batch mean:

$$y_{i,g}^{\Delta} = y_{i,g} - \mu_g(\mathbf{y}_{\cdot,g})$$

where $\mu_g(\cdot)$ denotes the batch mean of the expression values for the gene $g$ across $N$ spots and $y_{i,g}$ denotes the expression of the gene $g$ at spot $i$.

To form the predicted deviations, one can center the predicted gene expression values directly. Instead, we compute deviations in the embedding space, using $\mathbf{Z}$, to enhance more discriminative feature representations. We center each feature dimension across the batch and feed the centered features into the deviation prediction head $\rho_{dev}$ to predict gene-wise deviations:

$$\bar{\mathbf{y}}_i^{\Delta} = \rho_{dev}(\mathbf{z}_i^{\Delta}), \qquad z_{i,d}^{\Delta} = z_{i,d} - \bar{\mu}_d(\mathbf{z}_{\cdot,d})$$

where $z_{i,d}$ denotes the $d$-th embedding dimension of spot $i$ and $\bar{\mu}_d(\cdot)$ is the batch mean of the $d$-th embedding dimension across the $N$ spots.

Since the scale and range of gene expression differ across genes, we standardize deviations by normalizing with gene-wise standard deviations:

$$\tilde{y}_{i,g}^{\Delta} = \frac{y_{i,g}^{\Delta}}{\sigma_g(\mathbf{Y}^{\Delta}) + \epsilon}, \quad \tilde{\bar{y}}_{i,g}^{\Delta} = \frac{\bar{y}_{i,g}^{\Delta}}{\sigma_g(\bar{\mathbf{Y}}^{\Delta}) + \epsilon}$$

where $\sigma_g(\cdot)$ denotes the standard deviation of the gene $g$ and $\epsilon > 0$ is a small constant for numerical stability. The deviation-matching loss is then defined as

$$\mathcal{L}_{\mathrm{DEV}} = \frac{1}{NG} \sum_{i=1}^{N} \left\| \tilde{\mathbf{y}}_i^{\Delta} - \tilde{\bar{\mathbf{y}}}_i^{\Delta} \right\|_2^2.$$

#### 4.2.4. OVERALL OBJECTIVE

The full training objective is defined as

$$\mathcal{L} = \lambda_{\mathrm{MSE}}\mathcal{L}_{\mathrm{MSE}} + \lambda_{\mathrm{PL}}\mathcal{L}_{\mathrm{PL}} + \lambda_{\mathrm{TFA}}\mathcal{L}_{\mathrm{TFA}} + \lambda_{\mathrm{DEV}}\mathcal{L}_{\mathrm{DEV}}.$$

We set $\lambda_{\mathrm{MSE}} = 0.001$, $\lambda_{\mathrm{PL}} = 1.0$, $\lambda_{\mathrm{TFA}} = 0.1$, and $\lambda_{\mathrm{DEV}} = 0.1$.

## 5. Experiments

### 5.1. Gene Expression Prediction Datasets

We employed SpaRED (Mejia et al., 2024), a publicly available ST benchmark dataset to assess HEXST. SpaRED aggregates multiple ST cohorts spanning diverse tissues, species, and experimental protocols, and provides standardized preprocessing along with predefined train/validation/test splits to enable fair comparisons. Details of the dataset are provided in Appendix C.1.

### 5.2. Gene Expression Prediction Results

We compared HEXST with several state-of-the-art methods for spatial gene expression prediction: (1) STNet (He et al., 2020), (2) Hist2ST (Zeng et al., 2022), (3) EGNv1 (Yang et al., 2023), (4) TCGN (Xiao et al., 2024),

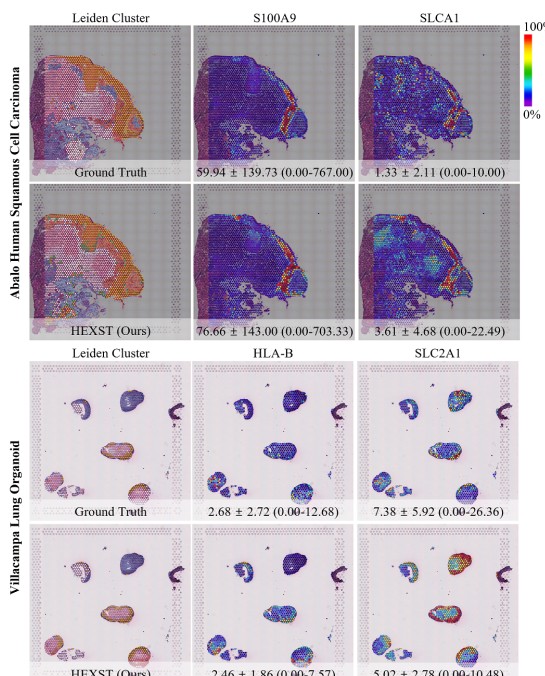

*Figure 4.* Qualitative comparison of spatial gene expression prediction results. Each column shows heatmaps of ground truth and HEXST-predicted gene expression for selected marker genes. Heatmap values are annotated as mean $\pm$ standard deviation (minimum–maximum) across spatial spots.

(5) EGNv2 (Yang et al., 2024), (6) NH2ST (Qu et al., 2025), and (7) PEKA (Pan et al., 2025). To provide a comprehensive evaluation, we employ five evaluation metrics: (1) $\mathrm{PCC_F}$: gene-wise PCC, (2) $\mathrm{PCC_S}$: spot-wise PCC, (3) $\mathrm{MI}_F$: gene-wise mutual information (MI), (4) $\mathrm{AUC_{0vNZ}}$: area under the curve (AUC) for distinguishing zero from non-zero expression, and (5) $\mathrm{AUC_{Q50}}$: AUC for discriminating between genes with expression levels above and below the median. Details of evaluation metrics are provided in Appendix C.2.

Table 1 presents the quantitative results for spatial gene expression prediction, averaged over the seven datasets from SpaRED. HEXST consistently achieved the strongest performance across all evaluation criteria. As compared with the best-performing baseline (PEKA), HEXST delivered consistent improvements of +0.0364 in $\mathrm{PCC_F}$, +0.0558 in $\mathrm{PCC_S}$, +0.0431 in $\mathrm{MI}_F$, +0.0312 in $\mathrm{AUC_{0vNZ}}$, and +0.0348 in $\mathrm{AUC_{Q50}}$, demonstrating robust performance gains at both the spot-, gene-, and dataset-level evaluations. Moreover, the investigation on each dataset confirmed the superior capability of HEXST. HEXST generally outperformed all baselines in the majority of datasets, ranking first in 5 out of 7 datasets for $\mathrm{PCC_F}$, $\mathrm{PCC_S}$, and $\mathrm{AUC_{0vNZ}}$, and 6 out of 7 datasets for $\mathrm{MI}_F$ and $\mathrm{AUC_{Q50}}$. In contrast, PEKA, the best-

*Table 1.* Comparison of gene expression prediction performance across different models. All values are averaged over seven SpaRED datasets, annotated as mean ± standard deviation. F and S denote gene-wise and spot-wise evaluations. Models are ordered by publication year. Best results are shown in **bold**, and second-best results are underlined.

| Model | $PCC_F \uparrow$ | $PCC_S \uparrow$ | $MI_F \uparrow$ | $AUC_{0vNZ} \uparrow$ | $AUC_{Q50} \uparrow$ |
|---|---|---|---|---|---|
| STNet (2020) | $0.0091 \pm 0.01$ | $0.7124 \pm 0.11$ | $0.0192 \pm 0.01$ | $0.4784 \pm 0.04$ | $0.5024 \pm 0.01$ |
| Hist2ST (2022) | $0.0516 \pm 0.00$ | $0.6599 \pm 0.09$ | $0.0788 \pm 0.03$ | $0.4933 \pm 0.02$ | $0.5000 \pm 0.00$ |
| EGNv1 (2023) | $0.0391 \pm 0.10$ | $0.6613 \pm 0.17$ | $0.0139 \pm 0.03$ | $0.5121 \pm 0.05$ | $0.5196 \pm 0.05$ |
| TCGN (2024) | $0.1920 \pm 0.26$ | $0.6740 \pm 0.64$ | $0.0628 \pm 0.08$ | $0.5571 \pm 0.61$ | $0.6066 \pm 0.65$ |
| EGNv2 (2024) | $0.2469 \pm 0.09$ | $0.5991 \pm 0.16$ | $0.0739 \pm 0.02$ | $0.6132 \pm 0.07$ | $0.6453 \pm 0.05$ |
| NH2ST (2025) | $0.3279 \pm 0.08$ | $0.7086 \pm 0.17$ | $0.1011 \pm 0.03$ | $0.6149 \pm 0.04$ | $0.6633 \pm 0.05$ |
| PEKA (2025) | $\underline{0.3863 \pm 0.07}$ | $\underline{0.7139 \pm 0.14}$ | $\underline{0.1109 \pm 0.02}$ | $\underline{0.6508 \pm 0.07}$ | $\underline{0.6879 \pm 0.03}$ |
| **HEXST (Ours)** | $\mathbf{0.4227 \pm 0.08}$ | $\mathbf{0.7697 \pm 0.12}$ | $\mathbf{0.1540 \pm 0.03}$ | $\mathbf{0.6820 \pm 0.04}$ | $\mathbf{0.7227 \pm 0.03}$ |

performing baseline on average, achieved the highest score for $AUC_{0vNZ}$ in VMB and $AUC_{Q50}$ in EHPCP1, but failed to match the consistent capability of HEXST across the full datasets and evaluation metrics. The complete results on all seven datasets are provided in Appendix C.3.

Qualitative assessments further highlight HEXST's ability to preserve gene-specific spatial contrasts (Fig. 4). Visual comparisons showed that the predicted expression heatmaps by HEXST closely aligned with the ground truth and effectively captured fine-grained spatial contrasts. In comparison to the best-performing baseline (PEKA), the strength of HEXST was further pronounced. While PEKA tended to blur the boundaries or mis-identified high-expression regions, HEXST retained sharp local variations and boundaries, exhibiting less over-smoothing. Additional qualitative results and visualizations are provided in Appendix C.4.

### 5.3. Ablation Study

We conducted a systematic ablation study to evaluate the individual and combined contributions of the proposed architectural components and four loss terms in HEXST, including $\mathcal{L}_{MSE}$, $\mathcal{L}_{PL}$, $\mathcal{L}_{TFA}$, and $\mathcal{L}_{DEV}$. The results, averaged over the seven SpaRED datasets, are summarized in Table 2.

For architectural ablations, we compared the full HEXST configuration, which uses a hexagonal window with HexRoPE, against two controlled variants: (i) a hexagonal window with 2D Cartesian RoPE and (ii) a square window with 2D Cartesian RoPE, while keeping all other settings unchanged. For the 2D Cartesian RoPE variants, relative $(x, y)$ positions were computed from the window center in the Cartesian coordinate system and then used to apply 2D RoPE. Under the full loss configuration, replacing the square window with the hexagonal window improves $PCC_F$ from 0.3690 to 0.3993, and introducing HexRoPE provides an additional gain of +0.0234 in $PCC_F$. These results confirm that the hexagonal window is essential for spatial neighborhood modeling, and HexRoPE provides additional gains through geometrically consistent positional information.

The absence of each loss term generally led to performance drops across most evaluation metrics, highlighting the synergistic effects of these terms. However, the impact of each term varied and metric-dependent. Specifically, the removal of $\mathcal{L}_{MSE}$ resulted in a substantial drop in spot-wise consistency with a decrease of 0.0446 in $PCC_S$. Omitting $\mathcal{L}_{PL}$ led to a reduction of 0.0686 in $PCC_F$, suggesting its role in preserving gene-wise predictions. Moreover, the removal of $\mathcal{L}_{TFA}$ and $\mathcal{L}_{DEV}$ decreased performance in most metrics, but provided improvements in specific areas ($MI_F$ and $AUC_{0vNZ}$). Removing $\mathcal{L}_{TFA}$ achieved the best $MI_F$ of 0.1615 and $AUC_{0vNZ}$ of 0.6933, while the absence of $\mathcal{L}_{DEV}$ resulted in the second-best $MI_F$ of 0.1611 and $AUC_{0vNZ}$ of 0.6892. These findings suggest that while $\mathcal{L}_{TFA}$ and $\mathcal{L}_{DEV}$ constrain the model to align with spatial variations in gene expression (improving PCC), relaxing these constraints allows the model to better capture global distribution (MI) and sparse patterns (AUC). Hence, the performance of HEXST can be further optimized through a tailored combination of these four loss terms.

### 5.4. Transcriptomics-guided Downstream Clinical Tasks on TCGA

To assess whether the transcriptomic representations learned by HEXST transfer to real-world clinical pathology tasks, we conducted two slide-level downstream evaluations on TCGA cohorts that lack spatial transcriptomics measurements: (1) prostate Gleason grading on TCGA-PRAD and (2) overall survival prediction on TCGA-LUAD. Details of the datasets are provided in Appendix D.1.

For both cohorts, we extracted visual features from WSIs and predicted gene expression vector (transcriptomics-guided spot representations) using HEXST and the best-performing baseline (PEKA). By aggregating visual features and transcriptomics-guided spot representations via multiple instance learning (MIL), we obtained slide-level representations, which were subsequently used to make slide-level predictions for both downstream tasks. For overall survival prediction, we included a Bulk RNA baseline that directly

*Table 2.* Ablation study of HEXST. We report performance for different architectural and loss configurations. Window and PE denote the spatial window shape and positional encoding, respectively. Loss configurations are indicated by O/X for each loss term. All metrics are higher-is-better. Best results are shown in **bold**, and second-best results are underlined.

| Window | PE | $\mathcal{L}_{\mathrm{MSE}}$ | $\mathcal{L}_{\mathrm{PL}}$ | $\mathcal{L}_{\mathrm{DEV}}$ | $\mathcal{L}_{\mathrm{TFA}}$ | $\mathrm{PCC_F}\uparrow$ | $\mathrm{PCC_S}\uparrow$ | $\mathrm{MI_F}\uparrow$ | $\mathrm{AUC_{0vNZ}}\uparrow$ | $\mathrm{AUC_{Q50}}\uparrow$ |
|---|---|---|---|---|---|---|---|---|---|---|
| Square | 2D RoPE | O | O | O | O | 0.3690±0.15 | 0.7421±0.09 | 0.1405±0.04 | 0.6617±0.04 | 0.6941±0.05 |
| Hexagonal | 2D RoPE | O | O | O | O | 0.3993±0.11 | 0.7508±0.09 | 0.1521±0.06 | 0.6670±0.03 | 0.7156±0.03 |
| Hexagonal | HexRoPE | O | X | O | O | 0.3541±0.13 | 0.7585±0.12 | 0.1350±0.04 | 0.6725±0.04 | 0.6980±0.04 |
| Hexagonal | HexRoPE | X | O | O | O | 0.3781±0.17 | 0.1194±0.13 | 0.1496±0.04 | 0.6721±0.04 | 0.7067±0.05 |
| Hexagonal | HexRoPE | O | O | X | X | 0.3972±0.15 | 0.6962±0.11 | 0.1498±0.06 | 0.6790±0.05 | 0.7159±0.05 |
| Hexagonal | HexRoPE | O | O | O | X | 0.3818±0.20 | 0.7375±0.09 | **0.1615±0.04** | **0.6933±0.04** | 0.7098±0.07 |
| Hexagonal | HexRoPE | O | O | X | O | 0.3796±0.19 | 0.7620±0.07 | 0.1611±0.03 | 0.6892±0.04 | 0.7094±0.06 |
| Hexagonal | HexRoPE | O | O | O | O | **0.4227±0.08** | **0.7697±0.12** | 0.1540±0.05 | 0.6820±0.05 | **0.7227±0.03** |

*Table 3.* Downstream clinical task performance on TCGA-PRAD Gleason grading (GG) and TCGA-LUAD overall survival (OS) prediction. Numbers represent mean ± standard deviation over five-fold cross-validation.

| Method | TCGA-PRAD GG | | TCGA-LUAD OS |
|---|---|---|---|
| | Acc. (%) ↑ | $\kappa$ ↑ | C-Index ↑ |
| Bulk RNA | - | - | 0.6042±0.05 |
| Image | 45.73±7.5 | 0.6405±0.06 | 0.5845±0.13 |
| Image + PEKA | 48.40±6.2 | 0.6473±0.05 | 0.5904±0.11 |
| Image + HEXST | 48.55±6.8 | 0.6743±0.09 | 0.5970±0.11 |

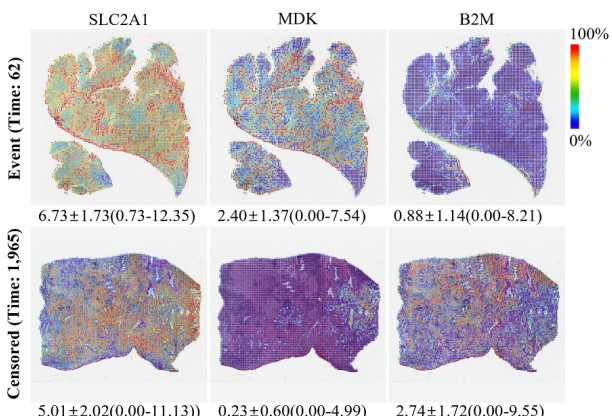

*Figure 5.* Qualitative examples of transcriptomics-guided survival prediction on TCGA-LUAD. A representative slide is shown, annotated with survival outcome (event vs. censored) and follow-up time (days), together with predicted spatial gene expression heatmaps. The numbers reported below each heatmap indicate the mean ± standard deviation with the minimum–maximum range of predicted expression computed across all spots within the slide.

leverages patient-level bulk RNA-seq profiles from TCGA, serving as a reference upper bound when transcriptomic measurements are directly available. For prostate cancer grading, two evaluation metrics were used: accuracy (ACC) and Cohen's $\kappa$. For overall survival prediction, the concordance index (C-index) was employed. Experimental details are available in Appendix D.2.

### 5.4.1. GLEASON GROUP CLASSIFICATION ON TCGA-PRAD

For prostate cancer grading on TCGA-PRAD, the transcriptomics-guided spot representations were obtained from the model trained on *Erickson Human Prostate Cancer P1* from SpaRED. The classification results showed that integrating transcriptomic signal substantially enhances predictive performance. Specifically, HEXST-driven transcriptomics-guided spot representations provided a performance gain of 2.82% in ACC and 0.0338 in $\kappa$ compared to the image-only baseline (Table 3). While PEKA-driven transcriptomics-guided spot representations also improved performance over the image-only baseline, the gains were notably smaller than those achieved by HEXST. This suggests that HEXST more effectively transfers transcriptomic signals that are relevant to prostate cancer grading.

### 5.4.2. SURVIVAL PREDICTION ON TCGA-LUAD

For survival analysis, we performed overall survival prediction on TCGA-LUAD. HEXST and PEKA, trained on the *Villacampa Lung Organoid* from SpaRED, were used to produce the transcriptomics-guided spot representations, which were then used to predict a patient-level risk score. The survival model was trained using the Cox proportional hazards loss (Cox, 1972). For bulk RNA-seq baseline, we used a Cox proportional hazards model. The experimental results demonstrate the clinical utility of the transcriptomics-guidance in survival prediction (Table 3). Compared to the image-only baseline, HEXST-driven transcriptomics-guided spot representations enhanced C-Index by 0.0115, whereas PEKA-driven transcriptomics-guided spot representations yielded a smaller improvement of 0.0059 in C-Index. Qualitative assessments further confirmed these findings, as

shown in Fig. 5. Although both HEXST and PEKA were able to provide clinically meaningful signals, they naturally fell short of the bulk RNA-seq baseline (difference of -0.0072 for HEXST and -0.0138 for PEKA). Nonetheless, HEXST approached the upper bound more closely than PEKA, underscoring its superior ability to infer prognostically relevant signals directly from histopathology images. Additional qualitative results and visualizations are provided in Appendix D.4.

## 6. Conclusion

In this work, we propose HEXST, a hexagon-aware transformer framework for predicting spatial gene expression from histopathology images. HEXST explicitly models the approximately hexagonal spot layouts commonly observed in ST platforms through multi-scale shifted hexagonal window attention and encodes non-Cartesian spatial relationships via HexRoPE. HEXST further leverages transcriptomic priors through feature alignment and a deviation-matching mechanism to mitigate over-smoothing and better preserve gene-specific spatial contrast. Across seven SpaRED datasets, HEXST consistently outperformed prior methods across comprehensive evaluation metrics. We also demonstrated transfer to two downstream tasks, including prostate cancer grading and overall survival prediction. These results highlight HEXST's potential for broader application in large histology-only clinical cohorts. Future work will compare scFoundation with recent cross-species or spatial foundation models, which can be incorporated into our framework without major architectural modifications by applying appropriate gene mapping according to each model's gene vocabulary. Future work will also extend HEXST to more diverse experimental settings, including broader organ types, larger target gene sets, additional spatial transcriptomics platforms, including continuous-coordinate spatial assays.

## Acknowledgement

This work was supported by a grant of the National Research Foundation of Korea (NRF) (No. RS-2025-00558322 and RS-2024-00397293) and the AI Computing Infrastructure Enhancement (GPU Rental Support) User Support Program funded by the Ministry of Science and ICT (MSIT) (No. RQT-25-120213), Republic of Korea.

## Impact Statement

This work addresses the problem of predicting spatial gene expression from histopathology images and highlights the potential to more broadly leverage spatial transcriptomic information without relying on costly experimental spatial transcriptomics assays. By enabling geometry-aware and gene-sensitive modeling of spatial expression patterns, our approach may facilitate large-scale exploratory studies of tissue organization and support transcriptomics-guided clinical research tasks such as prognosis prediction.

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

## A. Geometry Encoding and Hexagonal Window Construction

The geometry encoding and hexagonal window construction in HEXST are detailed in this section, including spot lattice scaling, axial coordinate conversion, multi-scale window center placement, and slot-based token packing.

**Hexagonal lattice representation**    We primarily represent hexagonal lattice locations using axial coordinates $(q, r) \in \mathbb{Z}^2$. In some cases, we convert the axial coordinates to cube coordinates $(u, v, w) \in \mathbb{Z}^3$:

$$(u, v, w) = (q, r, -q - r)$$

which satisfies $u + v + w = 0$. The distance between two points on a hexagonal lattice can be computed in the cube coordinates:

$$\mathcal{D}_{hex}((q, r), (q', r')) = \max(|\Delta q|, |\Delta r|, |\Delta q + \Delta r|)$$

where $\Delta q = q - q'$ and $\Delta r = r - r'$.

**Spot lattice scaling**    In a hexagonal lattice representation, the distance from the center to neighborhood points is assumed to be the same. However, the observed inter-spot distances in the ST data are not perfectly uniform in practice due to several factors such as tissue boundaries, missing spots, and coordinate noise. To obtain a regularized, robust representation, we estimate a representative spacing scale from the observed coordinates.

Given spot coordinates $\{\mathbf{x}_i\}_{i=1}^N$, we compute the median distance to the $k$-th nearest neighbor (we use $k{=}6$ to match the six-neighborhood structure in a hexagonal lattice):

$$d_{\mathrm{med}} = \mathrm{median}_i \left( \mathcal{D}_{Euc}(i, k) \right),$$

where $\mathcal{D}_{Euc}(i, k)$ denotes the Euclidean distance from spot $i$ to its $k$-th nearest neighbor. In the pointy-top hexagonal representation, the center-to-center distance between adjacent lattice points is $\sqrt{3}s$ where $s$ is the side length of a hexagon. Accordingly, we define the side length of the underlying spot hexagonal lattice as

$$s_{\mathrm{spot}} = \frac{d_{\mathrm{med}}}{\sqrt{3}}.$$

**Conversion from Cartesian to cube coordinates**    To convert Cartesian coordinates to pointy-top cube coordinates, we first express the coordinates relative to an anchor ($\mathbf{x}_{\mathrm{anchor}}$) (e.g., the first spot) and normalize them using $s_{\mathrm{spot}}$:

$$\tilde{\mathbf{x}}_i = \frac{\mathbf{x}_i - \mathbf{x}_{\mathrm{anchor}}}{s_{\mathrm{spot}}}.$$

We then map the normalized coordinates to fractional axial coordinates using a standard pointy-top conversion:

$$q_i^* = \frac{\sqrt{3}}{3}\tilde{\mathbf{x}}_i^{(1)} - \frac{1}{3}\tilde{\mathbf{x}}_i^{(2)}, \qquad r_i^* = \frac{2}{3}\tilde{\mathbf{x}}_i^{(2)},$$

where $(\tilde{\mathbf{x}}_i^{(1)}, \tilde{\mathbf{x}}_i^{(2)})$ are the Cartesian components of $\tilde{\mathbf{x}}_i$. The fractional axial coordinates are then converted to fractional cube coordinates as

$$(u_i^*, v_i^*, w_i^*) = (q_i^*, r_i^*, -q_i^* - r_i^*).$$

Finally, cube-rounding is applied to obtain integer cube coordinates:

$$(u_i, v_i, w_i) = \mathrm{cube\_round}(u_i^*, v_i^*, w_i^*),$$

where $\mathrm{cube\_round}(\cdot)$ denotes the nearest integer cube-rounding, which enforces the constraint $u + v + w = 0$.

**Multi-scale window centers via nearest-center assignment** Although a naive floor-division can define windows by binning lattice coordinates, it can introduce directional bias on hexagonal lattices and irregular boundary effect. Instead, HEXST places window centers on a coarser hexagonal lattice and assigns each spot to its nearest center, forming a Voronoi-style partition.

For stage $l$, let $K_l \in \mathbb{N}$ denote the window scale parameter. The center-to-center spacing of the window lattice is defined as

$$d^{(l)}_{\text{center}} = K_l \cdot d_{\text{med}},$$

and the corresponding hexagon side length as

$$s^{(l)}_{\text{center}} = \frac{d^{(l)}_{\text{center}}}{\sqrt{3}} = K_l \cdot s_{\text{spot}}.$$

Using $s^{(l)}_{\text{center}}$, we define the center-lattice basis vectors

$$\mathbf{e}^{(l)}_1 = (0, \sqrt{3}s^{(l)}_{\text{center}}), \qquad \mathbf{e}^{(l)}_2 = \left(\tfrac{3}{2}s^{(l)}_{\text{center}}, \tfrac{\sqrt{3}}{2}s^{(l)}_{\text{center}}\right).$$

Candidate window centers are generated by enumerating integer combinations

$$\mathbf{c}^{(l)}_{\alpha,\beta} = \mathbf{x}_{\text{anchor}} + \alpha\,\mathbf{e}^{(l)}_1 + \beta\,\mathbf{e}^{(l)}_2, \qquad \alpha, \beta \in \mathbb{Z}.$$

To promote cross-window information exchange, shifted windows are implemented by translating the entire set of centers by

$$\boldsymbol{\delta} \in \left\{\mathbf{0},\ \tfrac{1}{2}\mathbf{e}^{(l)}_1,\ \tfrac{1}{2}\mathbf{e}^{(l)}_2\right\},$$

which is applied across successive attention blocks within each stage. Each spot $i$ is assigned to its nearest (shifted) center:

$$b^{(l)}_i = \arg\min_b \|\mathbf{x}_i - (\mathbf{c}^{(l)}_b + \boldsymbol{\delta})\|^2_2,$$

where $b^{(l)}_i$ denotes the index of the window to which spot $i$ is assigned.

**Slot-based packing within each window** For efficient window-based self-attention with fixed tensor shapes, each window is represented using a shared set of discrete slots $\mathcal{S}_{K_l}$. The slot set is defined in cube coordinates as a hexagon of radius $K_l$ centered at the origin:

$$\mathcal{S}_{K_l} = \{(\Delta u, \Delta v, \Delta w) \in \mathbb{Z}^3 : \Delta u + \Delta v + \Delta w = 0,\ \max(|\Delta u|, |\Delta v|, |\Delta w|) \leq K_l\}.$$

Each element of $\mathcal{S}_{K_l}$ corresponds to a unique slot, and this slot ordering is shared across all windows at stage $l$.

Each window center $\mathbf{c}^{(l)}_b$ is mapped to cube coordinates $(u^{(l)}_b, v^{(l)}_b, w^{(l)}_b)$ using the same procedure as for spots. For a spot assigned to window $b^{(l)}_i$, the local cube-coordinate offset is computed as

$$(\Delta u^{(l)}_i, \Delta v^{(l)}_i, \Delta w^{(l)}_i) = (u_i - u^{(l)}_{b^{(l)}_i},\ v_i - v^{(l)}_{b^{(l)}_i},\ w_i - w^{(l)}_{b^{(l)}_i}).$$

By construction, each window only aggregates spots whose local offsets lie within $\mathcal{S}_{K_l}$; the corresponding spot features are written into the associated slots according to the predefined slot ordering.

This procedure yields a representation

$$\mathbf{H}^{(l)} \in \mathbb{R}^{B_l \times |\mathcal{S}_{K_l}| \times d},$$

Here, $B_l$ denotes the number of windows at stage $l$, $|\mathcal{S}_{K_l}|$ denotes the total number of discrete slots defined by the slot set $\mathcal{S}_{K_l}$, and $d$ denotes the feature dimension of each spot.

Table 4. Number of slides, spots, and target genes per dataset split.

| Dataset | Abbreviation | #Genes | #Slides | | | #Spots | | |
|---|---|---|---|---|---|---|---|---|
| | | | Train | Val | Test | Train | Val | Test |
| Abalo Human Squamous Cell Carcinoma (Abalo et al., 2021) | AHSCC | 128 | 2 | 1 | 1 | 5,542 | 2,496 | 2,336 |
| Erickson Human Prostate Cancer P1 (Erickson et al., 2022) | EHPCP1 | 128 | 4 | 2 | 1 | 12,479 | 5,298 | 2,972 |
| Mirzazadeh Mouse Bone (Mirzazadeh et al., 2023) | MMBO | 128 | 2 | 1 | 1 | 4,193 | 1,202 | 1,788 |
| Mirzazadeh Mouse Brain P1 (Mirzazadeh et al., 2023) | MMBP1 | 128 | 2 | 1 | 1 | 8,585 | 4,491 | 4,166 |
| Mirzazadeh Mouse Brain P2 (Mirzazadeh et al., 2023) | MMBP2 | 128 | 2 | 1 | 1 | 8,908 | 4,107 | 4,332 |
| Vicari Mouse Brain (Vicari et al., 2024) | VMB | 128 | 8 | 4 | 2 | 23,646 | 13,407 | 6,729 |
| Villacampa Lung Organoid (Villacampa et al., 2021) | VLO | 128 | 2 | 1 | 1 | 860 | 439 | 533 |

Table 5. Pairwise overlap of dataset-specific 128-gene panels.

| | AHSCC | EHPCP1 | MMBO | MMBP1 | MMBP2 | VMB | VLO |
|---|---|---|---|---|---|---|---|
| AHSCC | 128 | 17 | 6 | 3 | 4 | 5 | 14 |
| EHPCP1 | 17 | 128 | 6 | 4 | 5 | 8 | 21 |
| MMBO | 6 | 6 | 128 | 5 | 6 | 4 | 9 |
| MMBP1 | 3 | 4 | 5 | 128 | 64 | 53 | 4 |
| MMBP2 | 4 | 5 | 6 | 64 | 128 | 76 | 10 |
| VMB | 5 | 8 | 4 | 53 | 76 | 128 | 13 |
| VLO | 14 | 21 | 9 | 4 | 10 | 13 | 128 |

**Extension to irregular and continuous-coordinate assays** The core objective of HEXST is to enable more effective spatial context modeling. Among regular tessellations, hexagonal partition provides more uniform angular coverage through its 6-directional symmetry. This geometric property is universally applicable and is not limited to a specific coordinate system or data format.

To avoid hard coupling to a specific platform, HEXST uses a slot-based packing scheme. In this scheme, a fixed number of slots is defined within each window, and spots located inside the window are assigned to those predefined slots based on geometric distance. This scheme is not restricted to hexagonal grids and can naturally accommodate irregular inter-spot distances and missing spots within a window. The broad applicability of the proposed hexagonal window with slot-based packing scheme is supported by the downstream experiments on TCGA datasets. WSI data in the TCGA datasets follow a Cartesian grid, not a hexagonal grid. By utilizing the proposed slot-based packing scheme, the model trained solely on hexagonal grid-based data was directly applied to TCGA data without requiring architectural modifications.

By adjusting the coordinate normalization scheme, the same hexagonal windows can also be naturally defined in a continuous coordinate system. For example, the framework can be extended by partitioning the spatial space into hexagonal tiling while preserving continuous coordinates and assigning each point to its corresponding window, without requiring major modifications to the model architecture. In continuous coordinate settings, the slot packing within each hex window and the HexRoPE can be naturally extended from discrete formulations to continuous spatial environments.

## B. Implementation Details

Gene expression prediction experiments were conducted on a Linux server running Ubuntu 18.04.5 LTS, equipped with Intel Xeon Silver and 8 NVIDIA GeForce RTX 3090 GPUs. These experiments use Python 3.10 and PyTorch 2.6.

TCGA downstream experiments, including survival prediction and Gleason grading, are conducted on a separate Linux server running Ubuntu 24.04.2 LTS, equipped with Intel Xeon Platinum 8570 CPUs and a single NVIDIA H200 GPU. These experiments use Python 3.12 and PyTorch 2.8.

## C. Details on Spatial Gene Expression Prediction

### C.1. Dataset Details

All gene expression prediction experiments in this study are conducted on spatial transcriptomics datasets collected in SpaRED (Mejia et al., 2024). SpaRED aggregates Visium-based data across diverse tissues and species, and provides

*Table 6.* Overlap between dataset-specific 128-gene panels and the scFoundation vocabulary (19,264 genes).

| Dataset | Target | Matched | Missing | Missing genes |
|---|---|---|---|---|
| AHSCC | 128 | 122 | 6 | AES, H2AFV, IGHG1, IGKC, LOR, MALAT1 |
| EHPCP1 | 128 | 118 | 10 | C19ORF48, C1ORF21, H2AFJ, H2AFZ, MALAT1, PART1, SARS, SNHG19, SNHG25, SNHG8 |
| MMBO | 128 | 120 | 8 | 2310022B05RIK, CAR3, COX8B, HBA-A2, HBB-BS, HIST1H1A, MT1, SEPT5 |
| MMBP1 | 128 | 121 | 7 | 1110008P14RIK, 2010300C02RIK, CAR2, SCD2, SEPT4, SEPT5, TRF |
| MMBP2 | 128 | 120 | 8 | 1110008P14RIK, CAR2, CDR1OS, MT1, QK, SCD1, SCD2, TRF |
| VMB | 128 | 118 | 10 | 1110008P14RIK, CAR2, MALAT1, MEG3, OIP5OS1, PNMAL2, QK, SCD2, SEPT4, TRF |
| VLO | 128 | 122 | 6 | H1F0, MALAT1, MIAT, NORAD, SEPT2, SNHG25 |

predefined training, validation, and test splits to enable fair comparison. Each ST sample is provided at the slide level, where each slide consists of spot-level observations, and each spot is associated with an H&E image patch, a spatial coordinate, and a target gene expression vector. In this work, we directly use the datasets, target gene panels, and data splits released by SpaRED. Following the official protocol, we used the fixed train/val/test splits. We use seven datasets with an available test split, with a dataset-specific target panel of 128 genes. For each dataset, the 128 genes were independently selected from a larger gene set, with minimal overlaps among datasets (Table 5). This indicates that HEXST was not repeatedly evaluated on one shared gene set, but on multiple distinct transcriptomic subspaces.

Standard preprocessing steps are applied to the datasets, including count-based quality filtering, transcripts-per-million (TPM) normalization with log transformation, spatial smoothing, selection of spatially informative genes, and batch correction. We do not perform any additional filtering or preprocessing, and instead use the processed datasets as provided. The numbers of slides, spots, and target genes for each dataset are summarized in Table 4.

To examine the validity of transcriptomic feature alignment across species, we measured the overlap between each dataset-specific gene panel and the scFoundation vocabulary. Although scFoundation was originally trained on human genomic data and our experiments include mouse datasets, it is widely recognized that orthologous genes are conserved between humans and mice at the sequence, expression, and functional levels (Zheng-Bradley et al., 2010). Therefore, cross-species transcriptomic alignment remains biologically valid.

Examining the overlap between each dataset-specific gene panel and the scFoundation vocabulary 6, we found that 118–122 genes were matched across all datasets. Given this substantial overlap of orthologous genes and conserved gene expression patterns between humans and mice, scFoundation embeddings remain informative and provide sufficient signals for training HEXST, as supported by the consistent performance gains on the mouse datasets. Unmatched genes are mostly species-specific or non-coding RNAs (e.g., MALAT1, pseudogenes) excluded by scFoundation.

Importantly, while scFoundation is used for feature alignment during training, HEXST predicts all target genes. Since transcriptomic features use distributed representations, information from excluded genes can be indirectly captured through the matched genes.

### C.2. Evaluation Metrics

We evaluate the performance of spatial gene expression prediction using a comprehensive set of metrics that capture complementary aspects of prediction quality. Following prior work, we adopt the Pearson correlation coefficient (PCC) as the primary evaluation metric, and additionally report mutual information (MI), and area under the ROC curve (AUC). PCC measures the linear agreement between predicted and ground-truth gene expression values while being invariant to scale. MI measures nonlinear statistical dependency between predictions and ground truth. $AUC_{0vNZ}$ measures the ability to distinguish zero from non-zero gene expression values, $AUC_{Q50}$ evaluates discrimination between genes above and below the median expression level. Specifically, the notations F and S denote the following evaluation units. F (gene-wise) computes the correlation between predicted and ground-truth expression for each gene across all spatial locations and then averages the result over genes. S (spot-wise) computes the correlations between the genes for each individual spot.

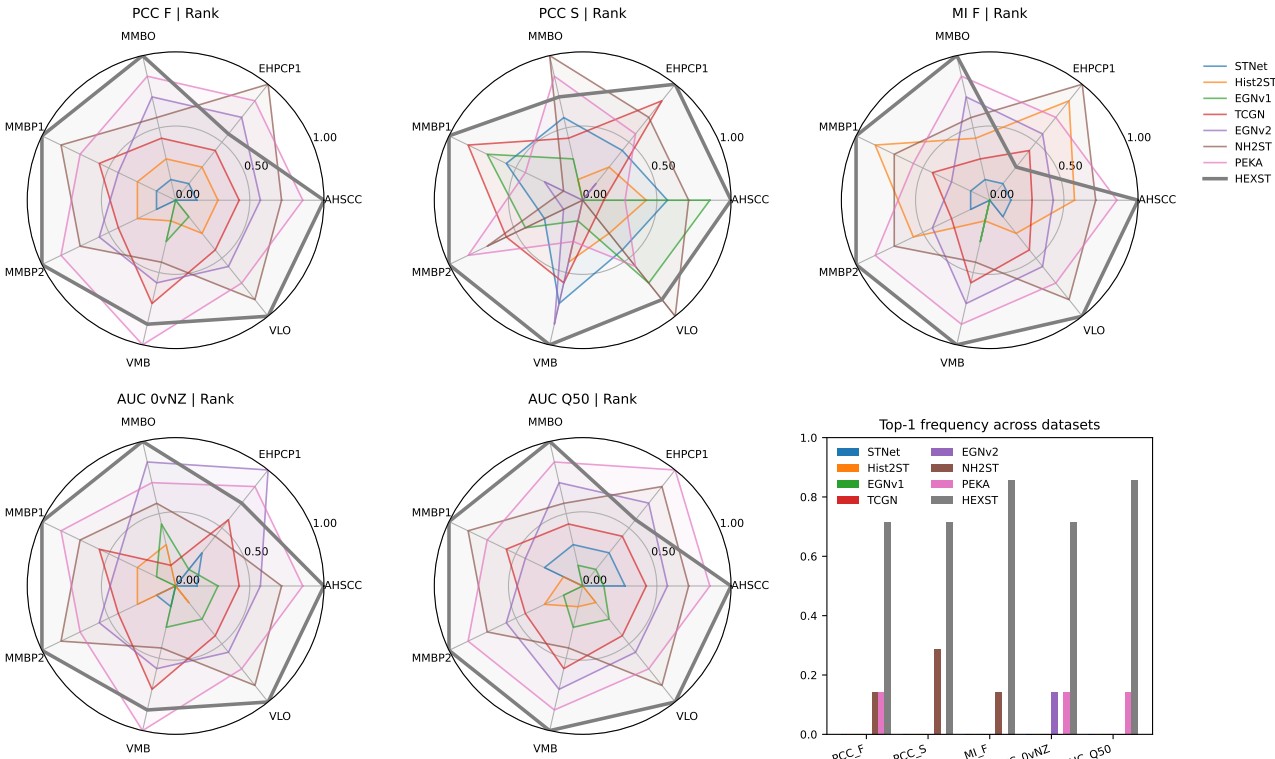

*Figure 6.* Rank-based comparison across seven SpaRED datasets over five metrics ($PCC_F$, $PCC_S$, $MI_F$, $AUC_{0vNZ}$, $AUC_{Q50}$). For each metric and dataset, models are ranked using dense ranking (1 = best), and the bar plot reports the *top-1 frequency* (fraction of datasets where a method attains rank 1).

### C.3. Per-Dataset Results on Seven Datasets

Table 7 reports full per-dataset gene expression prediction results on seven SpaRED datasets over five metrics, $PCC_F$, $PCC_S$, $MI_F$, $AUC_{0vNZ}$, and $AUC_{Q50}$. Figure 6 further summarizes this trend by visualizing rank-based comparisons aggregated over datasets, where HEXST attains the highest top-1 frequency for all metrics. Table 8 presents full per-dataset ablation results of HEXST, showing that combining the point-wise regression loss with the differential objective and transcriptomic feature alignment consistently yields the best $PCC_F$ and stable performance across datasets and metrics. Table 9 provides the per-dataset architectural ablation results under the full loss configuration.

### C.4. Qualitative Analysis

Figure 7 presents qualitative comparisons between ground-truth (GT) spatial gene expression and HEXST predictions on three datasets. For each dataset, we visualize Leiden clustering results and selected gene expression heatmaps.

In the Abalo Human Squamous Cell Carcinoma dataset, HEXST accurately recovers spatial domains that are well aligned with Leiden clusters, while faithfully reproducing expression patterns of representative marker genes such as *CD44*, *KRT5*, *KRT17*, *S100A9*, and *SLC2A1*. For example, *S100A9* is known to play an important role in tumor progression and tumor–microenvironment interactions, where elevated expression is associated with enhanced proliferation and invasion in solid tumors (Srikrishna, 2011). In addition, *SLC2A1* (encoding the glucose transporter GLUT1) is a key regulator of cancer metabolism, and its overexpression has been widely reported across multiple cancer types, including lung adenocarcinoma, where it is associated with aggressive tumor behavior and poor prognosis (Yu et al., 2017). These genes exhibit sharp spatial boundaries associated with tumor and epithelial regions, and such boundary structures are well preserved in the HEXST predictions. In the Erickson Human Prostate Cancer P1 dataset, HEXST predictions show strong agreement with both Leiden clusters and ground-truth expression patterns for prostate cancer–related genes including *CLPTM1L* and *FASN*. In

the Villacampa Lung Organoid dataset, immune- and metabolism-related spatial expression patterns are illustrated using *HLA-B* and *SLC2A1*. HEXST consistently reconstructs spatial regions associated with metabolic reprogramming or immune modulation, indicating that the predicted expression maps capture biologically meaningful spatial structures.

## D. Details on Downstream Clinical Tasks

### D.1. Dataset Details

### D.2. Experimental Settings

For each whole-slide image, we extract patch-level visual features using a pretrained UNI (Chen et al., 2024) pathology foundation model. In parallel, we apply a HEXST model trained on an external spatial transcriptomics (ST) dataset to predict spot-level gene expression from histology images. The predicted gene expression vectors are concatenated with the corresponding patch-level pathology features to form transcriptomics-guided spot representations. These transcriptomics-guided spot representations are then aggregated using CLAM (Lu et al., 2021), CLAM employs attention-based MIL with instance-level clustering regularization.

**Gleason Grading on TCGA-PRAD**  For Gleason grading on TCGA-PRAD, we use slide-level Gleason Grade Group (GG1–GG5) annotations provided in TCGA pathology records. When multiple diagnostic slides are available for a patient, a patient-level Gleason label is defined by majority voting over the slide-level GG annotations belonging to the same patient. Classification performance is evaluated using accuracy and Cohen's $\kappa$.

**Overall Survival Prediction on TCGA-LUAD**  For survival analysis on TCGA-LUAD, bulk RNA-seq profiles are used as input to a Cox proportional hazards (CoxPH) model (Cox, 1972). To reduce dimensionality and improve model stability, univariate gene filtering is applied using the training split only. Specifically, the concordance index (C-index) is computed independently for each gene, and genes with higher predictive power are selected. The resulting gene expression vectors are standardized using statistics computed on the training split, and a CoxPH model with an L2 regularization term is fitted. Survival prediction performance is evaluated using the concordance index (C-index) based on the predicted risk scores.

### D.3. Complete Results on Five Folds

Table 11 reports all fold-wise downstream performance over five-fold cross-validation for two clinical prediction tasks, TCGA-PRAD Gleason grading and TCGA-LUAD overall survival.

### D.4. Qualitative Analysis

Figure 8 illustrates how transcriptomics-guided predictions generated by HEXST on the TCGA-LUAD cohort reflect clinically meaningful heterogeneity in patient survival. For each representative patient, we visualize a slide thumbnail together with predicted spatial gene expression heatmaps for three prognostically relevant genes: *SLC2A1*, *MDK*, and *B2M*. All gene expression values shown in the figure are summarized as the mean ± standard deviation with the minimum–maximum range computed across spots within each slide, and the heatmap intensities are normalized using gene-wise minimum and maximum values estimated over the entire dataset.

Patients who experienced early events exhibit elevated spatial expression of *SLC2A1* and *MDK*. *SLC2A1* encodes the glucose transporter GLUT1 and is widely reported to be upregulated across multiple cancer types; in lung adenocarcinoma, its overexpression is closely associated with metabolic reprogramming and aggressive tumor behavior (Yu et al., 2017). Similarly, *MDK* (Midkine) is a growth factor known to be overexpressed in lung cancer, where increased expression has been linked to enhanced tumor growth, invasion, and early postoperative recurrence (Yang et al., 2025). Concurrently, *B2M* expression in these patients is reduced or exhibits spatially heterogeneous patterns. *B2M* (beta-2 microglobulin) is a core component of the MHC class I complex required for antigen presentation to CD8$^{+}$ T cells, and its loss or downregulation has been shown to impair anti-tumor immune recognition and contribute to resistance to immunotherapy in lung adenocarcinoma (Miao et al., 2018).

In contrast, censored patients with longer follow-up times tend to show relatively lower expression of oncogenic and metabolic markers such as *SLC2A1* and *MDK*, while exhibiting better preserved expression of immune-related genes such as *B2M*. Notably, *B2M* expression in censored cases is generally higher than that observed in patients who experienced early

events.

Importantly, these differences are not confined to global averages but manifest as spatially structured patterns across the tissue. HEXST successfully reconstructs such intra-slide heterogeneity, capturing localized regions of elevated metabolic or proliferative activity as well as immune-depleted niches. These qualitative observations complement the quantitative survival prediction results and support that transcriptomics-guided representations learned by HEXST encode clinically relevant biological signals that generalize to large histology-only cohorts lacking spatial transcriptomics measurements.

*Table 7.* Per-dataset gene expression prediction performance across seven spatial transcriptomics datasets.

**(A) AHSCC (Abalo Human Squamous Cell Carcinoma)**

| Model | $PCC_F$ | $PCC_S$ | $MI_F$ | $AUC_{0vNZ}$ | $AUC_{Q50}$ |
|---|---|---|---|---|---|
| STNet | 0.0266 | 0.6603 | 0.0263 | 0.4993 | 0.5101 |
| Hist2ST | 0.0515 | 0.6250 | 0.0827 | 0.4982 | 0.4988 |
| EGNv1 | 0.0006 | 0.6775 | 0.0021 | 0.5005 | 0.4998 |
| TCGN | 0.2577 | 0.3812 | 0.0535 | 0.5535 | 0.6110 |
| EGNv2 | 0.3008 | 0.3538 | 0.0662 | 0.6215 | 0.6583 |
| NH2ST | 0.3701 | 0.6701 | 0.0925 | 0.6250 | 0.6660 |
| PEKA | 0.4884 | 0.6025 | 0.1433 | 0.6522 | 0.7068 |
| HEXST | 0.5643 | 0.7363 | 0.1980 | 0.6996 | 0.7505 |

**(B) EHPCP1 (Erickson Human Prostate Cancer P1)**

| Model | $PCC_F$ | $PCC_S$ | $MI_F$ | $AUC_{0vNZ}$ | $AUC_{Q50}$ |
|---|---|---|---|---|---|
| STNet | 0.0107 | 0.5508 | 0.0250 | 0.5075 | 0.5054 |
| Hist2ST | 0.0457 | 0.5502 | 0.1189 | 0.4860 | 0.4963 |
| EGNv1 | 0.0000 | 0.3402 | 0.0032 | 0.5002 | 0.4999 |
| TCGN | 0.1445 | 0.5909 | 0.0804 | 0.5496 | 0.6300 |
| EGNv2 | 0.2048 | 0.4244 | 0.0895 | 0.6668 | 0.6641 |
| NH2ST | 0.3821 | 0.5668 | 0.1413 | 0.5221 | 0.6687 |
| PEKA | 0.3059 | 0.5667 | 0.0920 | 0.6366 | 0.6878 |
| HEXST | 0.1600 | 0.6271 | 0.0754 | 0.5877 | 0.6454 |

**(C) MMBO (Mirzazadeh Mouse Bone)**

| Model | $PCC_F$ | $PCC_S$ | $MI_F$ | $AUC_{0vNZ}$ | $AUC_{Q50}$ |
|---|---|---|---|---|---|
| STNet | 0.0127 | 0.7724 | 0.0177 | 0.4136 | 0.5029 |
| Hist2ST | 0.0466 | 0.6521 | 0.0816 | 0.4568 | 0.4959 |
| EGNv1 | 0.0000 | 0.6581 | 0.0064 | 0.4648 | 0.4997 |
| TCGN | 0.0791 | 0.7712 | 0.0429 | 0.4528 | 0.5403 |
| EGNv2 | 0.3605 | 0.5348 | 0.1125 | 0.5843 | 0.6946 |
| NH2ST | 0.2062 | 0.8200 | 0.0930 | 0.5053 | 0.6068 |
| PEKA | 0.4589 | 0.8051 | 0.1341 | 0.5815 | 0.6954 |
| HEXST | 0.5119 | 0.8027 | 0.2266 | 0.6531 | 0.7548 |

**(D) MMBP1 (Mirzazadeh Mouse Brain P1)**

| Model | $PCC_F$ | $PCC_S$ | $MI_F$ | $AUC_{0vNZ}$ | $AUC_{Q50}$ |
|---|---|---|---|---|---|
| STNet | 0.0251 | 0.7943 | 0.0153 | 0.4891 | 0.5115 |
| Hist2ST | 0.0562 | 0.6978 | 0.1024 | 0.5081 | 0.5040 |
| EGNv1 | 0.0016 | 0.7945 | 0.0040 | 0.5041 | 0.5000 |
| TCGN | 0.1117 | 0.8094 | 0.0527 | 0.5661 | 0.5701 |
| EGNv2 | 0.1070 | 0.7899 | 0.0250 | 0.5596 | 0.5546 |
| NH2ST | 0.2882 | 0.7582 | 0.0763 | 0.6685 | 0.6474 |
| PEKA | 0.2740 | 0.7917 | 0.0622 | 0.6740 | 0.6288 |
| HEXST | 0.4240 | 0.8415 | 0.1476 | 0.7599 | 0.7211 |

**(E) MMBP2 (Mirzazadeh Mouse Brain P2)**

| Model | $PCC_F$ | $PCC_S$ | $MI_F$ | $AUC_{0vNZ}$ | $AUC_{Q50}$ |
|---|---|---|---|---|---|
| STNet | 0.0041 | 0.6988 | 0.0133 | 0.5003 | 0.4979 |
| Hist2ST | 0.0556 | 0.6469 | 0.0679 | 0.5036 | 0.5048 |
| EGNv1 | 0.0000 | 0.7015 | 0.0022 | 0.5001 | 0.5001 |
| TCGN | 0.2238 | 0.7254 | 0.0459 | 0.5730 | 0.6058 |
| EGNv2 | 0.2295 | 0.6567 | 0.0485 | 0.6139 | 0.6321 |
| NH2ST | 0.3580 | 0.7515 | 0.1065 | 0.6766 | 0.6953 |
| PEKA | 0.4126 | 0.7516 | 0.1099 | 0.6702 | 0.7010 |
| HEXST | 0.4630 | 0.7842 | 0.1562 | 0.7094 | 0.7429 |

**(F) VMB (Vicari Mouse Brain)**

| Model | $PCC_F$ | $PCC_S$ | $MI_F$ | $AUC_{0vNZ}$ | $AUC_{Q50}$ |
|---|---|---|---|---|---|
| STNet | 0.0000 | 0.6307 | 0.0144 | 0.5047 | 0.4979 |
| Hist2ST | 0.0553 | 0.6036 | 0.0713 | 0.5035 | 0.5021 |
| EGNv1 | 0.2714 | 0.5733 | 0.0715 | 0.6135 | 0.6368 |
| TCGN | 0.3257 | 0.6202 | 0.1097 | 0.6829 | 0.6842 |
| EGNv2 | 0.2971 | 0.6501 | 0.1194 | 0.6679 | 0.6848 |
| NH2ST | 0.2925 | 0.4942 | 0.0868 | 0.6573 | 0.6530 |
| PEKA | 0.4402 | 0.5986 | 0.1525 | 0.7123 | 0.7295 |
| HEXST | 0.4330 | 0.7040 | 0.1560 | 0.7069 | 0.7332 |

**(G) VLO (Villacampa Lung Organoid)**

| Model | $PCC_F$ | $PCC_S$ | $MI_F$ | $AUC_{0vNZ}$ | $AUC_{Q50}$ |
|---|---|---|---|---|---|
| STNet | -0.0154 | 0.8795 | 0.0219 | 0.4340 | 0.4907 |
| Hist2ST | 0.0505 | 0.8440 | 0.0266 | 0.4971 | 0.4978 |
| EGNv1 | 0.0000 | 0.8838 | 0.0078 | 0.5013 | 0.5010 |
| TCGN | 0.2013 | 0.8197 | 0.0547 | 0.5220 | 0.6049 |
| EGNv2 | 0.2287 | 0.7841 | 0.0560 | 0.5784 | 0.6283 |
| NH2ST | 0.3980 | 0.8990 | 0.1114 | 0.6496 | 0.7061 |
| PEKA | 0.3238 | 0.8812 | 0.0820 | 0.6291 | 0.6658 |
| HEXST | 0.4026 | 0.8919 | 0.1183 | 0.6577 | 0.7112 |

*Table 8.* Per-dataset ablation results of HEXST across spatial transcriptomics datasets. We report performance for different loss configurations, indicated by ○/✗ for each loss term. All metrics are higher-is-better.

**(A) AHSCC (Abalo Human Squamous Cell Carcinoma)**

| $\mathcal{L}_{MSE}$ | $\mathcal{L}_{PL}$ | $\mathcal{L}_{DEV}$ | $\mathcal{L}_{TFA}$ | $PCC_F$ | $PCC_S$ | $MI_F$ | $AUC_{0vNZ}$ | $AUC_{Q50}$ |
|---|---|---|---|---|---|---|---|---|
| X | O | O | O | 0.5333 | 0.0574 | 0.1816 | 0.6879 | 0.7484 |
| O | X | O | O | 0.4904 | 0.7425 | 0.1580 | 0.6635 | 0.7148 |
| O | O | X | X | 0.5502 | 0.7206 | 0.1927 | 0.6992 | 0.7511 |
| O | O | X | O | 0.5561 | 0.7274 | 0.1923 | 0.6961 | 0.7513 |
| O | O | O | X | 0.5354 | 0.7153 | 0.1869 | 0.6968 | 0.7476 |
| O | O | O | O | 0.5643 | 0.7363 | 0.1980 | 0.6996 | 0.7505 |

**(B) EHPCP1 (Erickson Human Prostate Cancer P1)**

| $\mathcal{L}_{MSE}$ | $\mathcal{L}_{PL}$ | $\mathcal{L}_{DEV}$ | $\mathcal{L}_{TFA}$ | $PCC_F$ | $PCC_S$ | $MI_F$ | $AUC_{0vNZ}$ | $AUC_{Q50}$ |
|---|---|---|---|---|---|---|---|---|
| X | O | O | O | -0.0019 | 0.1390 | 0.0926 | 0.6454 | 0.5911 |
| O | X | O | O | 0.0628 | 0.5470 | 0.0746 | 0.6134 | 0.6164 |
| O | O | X | X | 0.0704 | 0.5751 | 0.0503 | 0.5851 | 0.6165 |
| O | O | X | O | -0.0551 | 0.6348 | 0.1215 | 0.6415 | 0.5588 |
| O | O | O | X | -0.0914 | 0.5738 | 0.1177 | 0.6355 | 0.5446 |
| O | O | O | O | 0.1600 | 0.6271 | 0.0754 | 0.5877 | 0.6454 |

**(C) MMBO (Mirzazadeh Mouse Bone)**

| $\mathcal{L}_{MSE}$ | $\mathcal{L}_{PL}$ | $\mathcal{L}_{DEV}$ | $\mathcal{L}_{TFA}$ | $PCC_F$ | $PCC_S$ | $MI_F$ | $AUC_{0vNZ}$ | $AUC_{Q50}$ |
|---|---|---|---|---|---|---|---|---|
| X | O | O | O | 0.5437 | 0.2791 | 0.2176 | 0.5909 | 0.7429 |
| O | X | O | O | 0.4537 | 0.7834 | 0.2064 | 0.6402 | 0.7361 |
| O | O | X | X | 0.5331 | 0.8259 | 0.2413 | 0.6732 | 0.7736 |
| O | O | X | O | 0.4964 | 0.8025 | 0.2207 | 0.6563 | 0.7606 |
| O | O | O | X | 0.5149 | 0.8157 | 0.2470 | 0.6851 | 0.7786 |
| O | O | O | O | 0.5119 | 0.8027 | 0.2266 | 0.6531 | 0.7548 |

**(D) MMBP1 (Mirzazadeh Mouse Brain P1)**

| $\mathcal{L}_{MSE}$ | $\mathcal{L}_{PL}$ | $\mathcal{L}_{DEV}$ | $\mathcal{L}_{TFA}$ | $PCC_F$ | $PCC_S$ | $MI_F$ | $AUC_{0vNZ}$ | $AUC_{Q50}$ |
|---|---|---|---|---|---|---|---|---|
| X | O | O | O | 0.3774 | 0.0221 | 0.1357 | 0.7220 | 0.7045 |
| O | X | O | O | 0.3494 | 0.8286 | 0.1282 | 0.7478 | 0.6887 |
| O | O | X | X | 0.3909 | 0.7615 | 0.1360 | 0.7361 | 0.7020 |
| O | O | X | O | 0.3916 | 0.8579 | 0.1464 | 0.7641 | 0.7092 |
| O | O | O | X | 0.4253 | 0.8530 | 0.1537 | 0.7680 | 0.7185 |
| O | O | O | O | 0.4240 | 0.8415 | 0.1476 | 0.7599 | 0.7211 |

**(E) MMBP2 (Mirzazadeh Mouse Brain P2)**

| $\mathcal{L}_{MSE}$ | $\mathcal{L}_{PL}$ | $\mathcal{L}_{DEV}$ | $\mathcal{L}_{TFA}$ | $PCC_F$ | $PCC_S$ | $MI_F$ | $AUC_{0vNZ}$ | $AUC_{Q50}$ |
|---|---|---|---|---|---|---|---|---|
| X | O | O | O | 0.4596 | 0.1672 | 0.1592 | 0.7129 | 0.7428 |
| O | X | O | O | 0.4164 | 0.7767 | 0.1335 | 0.6929 | 0.7213 |
| O | O | X | X | 0.4672 | 0.7935 | 0.1575 | 0.7128 | 0.7438 |
| O | O | X | O | 0.4696 | 0.7942 | 0.1624 | 0.7121 | 0.7465 |
| O | O | O | X | 0.4776 | 0.7915 | 0.1608 | 0.7107 | 0.7450 |
| O | O | O | O | 0.4630 | 0.7842 | 0.1562 | 0.7094 | 0.7429 |

**(F) VMB (Vicari Mouse Brain)**

| $\mathcal{L}_{MSE}$ | $\mathcal{L}_{PL}$ | $\mathcal{L}_{DEV}$ | $\mathcal{L}_{TFA}$ | $PCC_F$ | $PCC_S$ | $MI_F$ | $AUC_{0vNZ}$ | $AUC_{Q50}$ |
|---|---|---|---|---|---|---|---|---|
| X | O | O | O | 0.4042 | 0.2823 | 0.1577 | 0.6981 | 0.7245 |
| O | X | O | O | 0.3748 | 0.6708 | 0.1468 | 0.7064 | 0.7188 |
| O | O | X | X | 0.4271 | 0.7000 | 0.1590 | 0.7046 | 0.7321 |
| O | O | X | O | 0.4580 | 0.6928 | 0.1680 | 0.7059 | 0.7399 |
| O | O | O | X | 0.4604 | 0.7060 | 0.1640 | 0.7137 | 0.7410 |
| O | O | O | O | 0.4330 | 0.7040 | 0.1560 | 0.7069 | 0.7332 |

**(G) VLO (Villacampa Lung Organoid)**

| $\mathcal{L}_{MSE}$ | $\mathcal{L}_{PL}$ | $\mathcal{L}_{DEV}$ | $\mathcal{L}_{TFA}$ | $PCC_F$ | $PCC_S$ | $MI_F$ | $AUC_{0vNZ}$ | $AUC_{Q50}$ |
|---|---|---|---|---|---|---|---|---|
| X | O | O | O | 0.3302 | -0.1112 | 0.1028 | 0.6474 | 0.6928 |
| O | X | O | O | 0.3312 | 0.9606 | 0.0971 | 0.6430 | 0.6901 |
| O | O | X | X | 0.3418 | 0.4969 | 0.1116 | 0.6418 | 0.6919 |
| O | O | X | O | 0.3405 | 0.8241 | 0.1163 | 0.6487 | 0.6993 |
| O | O | O | X | 0.3503 | 0.7070 | 0.1003 | 0.6437 | 0.6931 |
| O | O | O | O | 0.4026 | 0.8919 | 0.1183 | 0.6577 | 0.7112 |

*Table 9.* Per-dataset architectural ablation results of HEXST across spatial transcriptomics datasets. We report performance for different window shapes and positional encoding methods. All metrics are higher-is-better.

**(A) AHSCC (Abalo Human Squamous Cell Carcinoma)**

| Window | PE | $PCC_F$ | $PCC_S$ | $MI_F$ | $AUC_{0vNZ}$ | $AUC_{Q50}$ |
|---|---|---|---|---|---|---|
| Square | 2D RoPE | 0.5097 | 0.7020 | 0.1670 | 0.6702 | 0.7305 |
| Hexagonal | 2D RoPE | 0.5357 | 0.7300 | 0.1872 | 0.6833 | 0.7403 |
| Hexagonal | HexRoPE | 0.5643 | 0.7363 | 0.1980 | 0.6996 | 0.7505 |

**(B) EHPCP1 (Erickson Human Prostate Cancer P1)**

| Window | PE | $PCC_F$ | $PCC_S$ | $MI_F$ | $AUC_{0vNZ}$ | $AUC_{Q50}$ |
|---|---|---|---|---|---|---|
| Square | 2D RoPE | 0.0341 | 0.6381 | 0.0891 | 0.6381 | 0.5900 |
| Hexagonal | 2D RoPE | 0.1837 | 0.5930 | 0.0743 | 0.6184 | 0.6538 |
| Hexagonal | HexRoPE | 0.1600 | 0.6271 | 0.0754 | 0.5877 | 0.6454 |

**(C) MMBO (Mirzazadeh Mouse Bone)**

| Window | PE | $PCC_F$ | $PCC_S$ | $MI_F$ | $AUC_{0vNZ}$ | $AUC_{Q50}$ |
|---|---|---|---|---|---|---|
| Square | 2D RoPE | 0.4826 | 0.7642 | 0.2112 | 0.5735 | 0.7071 |
| Hexagonal | 2D RoPE | 0.5102 | 0.7492 | 0.2701 | 0.6887 | 0.7640 |
| Hexagonal | HexRoPE | 0.5119 | 0.8027 | 0.2266 | 0.6531 | 0.7548 |

**(D) MMBP1 (Mirzazadeh Mouse Brain P1)**

| Window | PE | $PCC_F$ | $PCC_S$ | $MI_F$ | $AUC_{0vNZ}$ | $AUC_{Q50}$ |
|---|---|---|---|---|---|---|
| Square | 2D RoPE | 0.3485 | 0.7867 | 0.1016 | 0.7057 | 0.6719 |
| Hexagonal | 2D RoPE | 0.3584 | 0.8317 | 0.1413 | 0.6458 | 0.7026 |
| Hexagonal | HexRoPE | 0.4240 | 0.8415 | 0.1476 | 0.7599 | 0.7211 |

**(E) MMBP2 (Mirzazadeh Mouse Brain P2)**

| Window | PE | $PCC_F$ | $PCC_S$ | $MI_F$ | $AUC_{0vNZ}$ | $AUC_{Q50}$ |
|---|---|---|---|---|---|---|
| Square | 2D RoPE | 0.4284 | 0.7822 | 0.1481 | 0.6967 | 0.7304 |
| Hexagonal | 2D RoPE | 0.4221 | 0.7842 | 0.1378 | 0.6954 | 0.7291 |
| Hexagonal | HexRoPE | 0.4630 | 0.7842 | 0.1562 | 0.7094 | 0.7429 |

**(F) VMB (Vicari Mouse Brain)**

| Window | PE | $PCC_F$ | $PCC_S$ | $MI_F$ | $AUC_{0vNZ}$ | $AUC_{Q50}$ |
|---|---|---|---|---|---|---|
| Square | 2D RoPE | 0.3668 | 0.6217 | 0.1461 | 0.7008 | 0.7144 |
| Hexagonal | 2D RoPE | 0.3996 | 0.6663 | 0.1429 | 0.6973 | 0.7158 |
| Hexagonal | HexRoPE | 0.4330 | 0.7040 | 0.1560 | 0.7069 | 0.7332 |

**(G) VLO (Villacampa Lung Organoid)**

| Window | PE | $PCC_F$ | $PCC_S$ | $MI_F$ | $AUC_{0vNZ}$ | $AUC_{Q50}$ |
|---|---|---|---|---|---|---|
| Square | 2D RoPE | 0.4128 | 0.9003 | 0.1203 | 0.6468 | 0.7141 |
| Hexagonal | 2D RoPE | 0.3856 | 0.9012 | 0.1111 | 0.6402 | 0.7033 |
| Hexagonal | HexRoPE | 0.4026 | 0.8919 | 0.1183 | 0.6577 | 0.7112 |

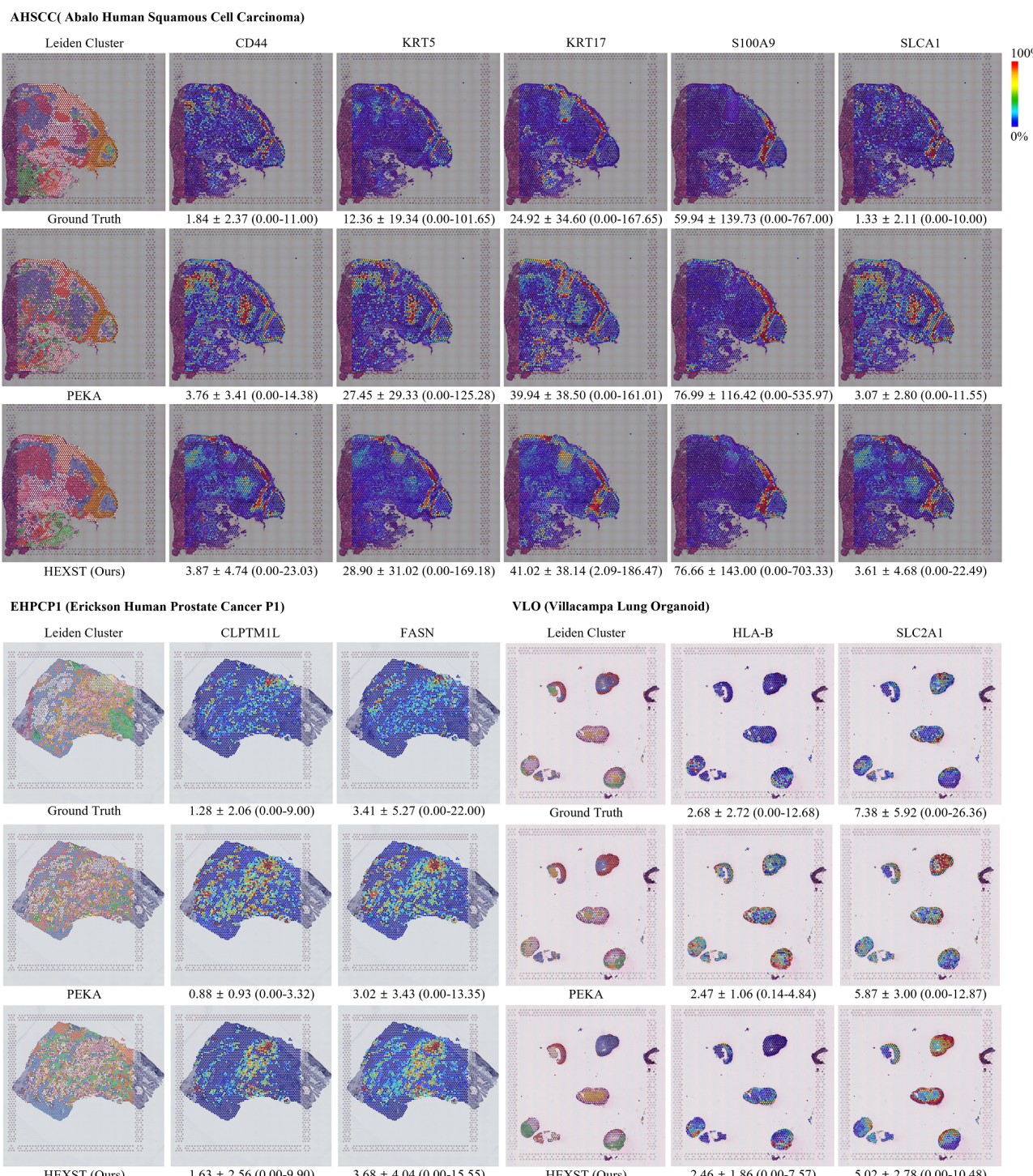

*Figure 7.* Qualitative comparison of spatial gene expression prediction results. For each dataset, the first column shows Leiden clustering results, followed by ground-truth (GT) and HEXST-predicted gene expression heatmaps for selected marker genes. Results are shown for Abalo Human Squamous Cell Carcinoma, Erickson Human Prostate Cancer P1, and Villacampa Lung Organoid. Heatmap values are annotated as mean ± standard deviation (min–max) across spatial spots.

*Table 10.* Fold-wise test set statistics for two clinical prediction tasks. **(A)** TCGA-PRAD Gleason Group (GG1–GG5) classification and **(B)** TCGA-LUAD Overall Survival (OS) prediction. In all entries, we report the number of patients with the corresponding number of slides shown in parentheses. For survival analysis, follow-up time is summarized using median and standard deviation (in days), stratified by event and censoring status.

| **(A) TCGA-PRAD: Gleason Group classification** | | | | | | **(B) TCGA-LUAD: Overall Survival (OS) prediction** | | | | |
|---|---|---|---|---|---|---|---|---|---|---|
| | #Patients (#Slides) | | | | | | #Patients (#Slides) | | Time: Med±Std | |
| Fold | GG1 | GG2 | GG3 | GG4 | GG5 | Fold | Event | Censored | Event | Censored |
| F0 | 7 (9) | 21 (21) | 15 (15) | 9 (14) | 18 (19) | F0 | 33 (34) | 59 (70) | $626 \pm 450$ | $690 \pm 777$ |
| F1 | 6 (6) | 21 (21) | 14 (14) | 10 (10) | 18 (24) | F1 | 33 (50) | 58 (58) | $598 \pm 525$ | $701 \pm 1349$ |
| F2 | 6 (6) | 21 (21) | 15 (15) | 9 (16) | 18 (20) | F2 | 32 (34) | 59 (63) | $624 \pm 579$ | $691 \pm 928$ |
| F3 | 6 (6) | 21 (21) | 15 (15) | 9 (9) | 18 (28) | F3 | 32 (47) | 59 (63) | $642 \pm 840$ | $704 \pm 1037$ |
| F4 | 6 (6) | 21 (23) | 15 (18) | 9 (9) | 18 (19) | F4 | 32 (41) | 59 (59) | $711 \pm 539$ | $683 \pm 822$ |
| TOTAL | 31 (33) | 105 (107) | 74 (77) | 46 (58) | 90 (110) | TOTAL | 162 (206) | 294 (313) | $626 \pm 597$ | $691 \pm 999$ |

*Table 11.* Fold-wise downstream performance over five-fold cross-validation. Results are reported for TCGA-PRAD Gleason Grading (accuracy and Cohen's $\kappa$) and TCGA-LUAD overall survival (C-index).

| Method | Fold | TCGA-PRAD Gleason Grading | | TCGA-LUAD Overall Survival |
|---|---|---|---|---|
| | | Acc. (%) ↑ | $\kappa$ ↑ | C-Index ↑ |
| Bulk RNA | F1 | 36.36 | 0.1762 | 0.5839 |
| | F2 | 50.67 | 0.3575 | 0.6005 |
| | F3 | 39.74 | 0.2282 | 0.5423 |
| | F4 | 32.91 | 0.1277 | 0.6356 |
| | F5 | 31.08 | 0.0963 | 0.6588 |
| Image | F1 | 41.03 | 0.5773 | 0.5019 |
| | F2 | 52.00 | 0.6697 | 0.7134 |
| | F3 | 55.13 | 0.7258 | 0.3949 |
| | F4 | 40.51 | 0.6251 | 0.6635 |
| | F5 | 40.00 | 0.6044 | 0.6488 |
| Image + PEKA | F1 | 44.16 | 0.5851 | 0.5034 |
| | F2 | 54.67 | 0.6759 | 0.6990 |
| | F3 | 55.13 | 0.7028 | 0.4409 |
| | F4 | 49.37 | 0.6693 | 0.6580 |
| | F5 | 38.67 | 0.6035 | 0.6508 |
| Image + HEXST | F1 | 46.15 | 0.5704 | 0.5077 |
| | F2 | 56.00 | 0.7503 | 0.7047 |
| | F3 | 53.85 | 0.7795 | 0.4551 |
| | F4 | 48.10 | 0.6729 | 0.6635 |
| | F5 | 38.67 | 0.5983 | 0.6538 |

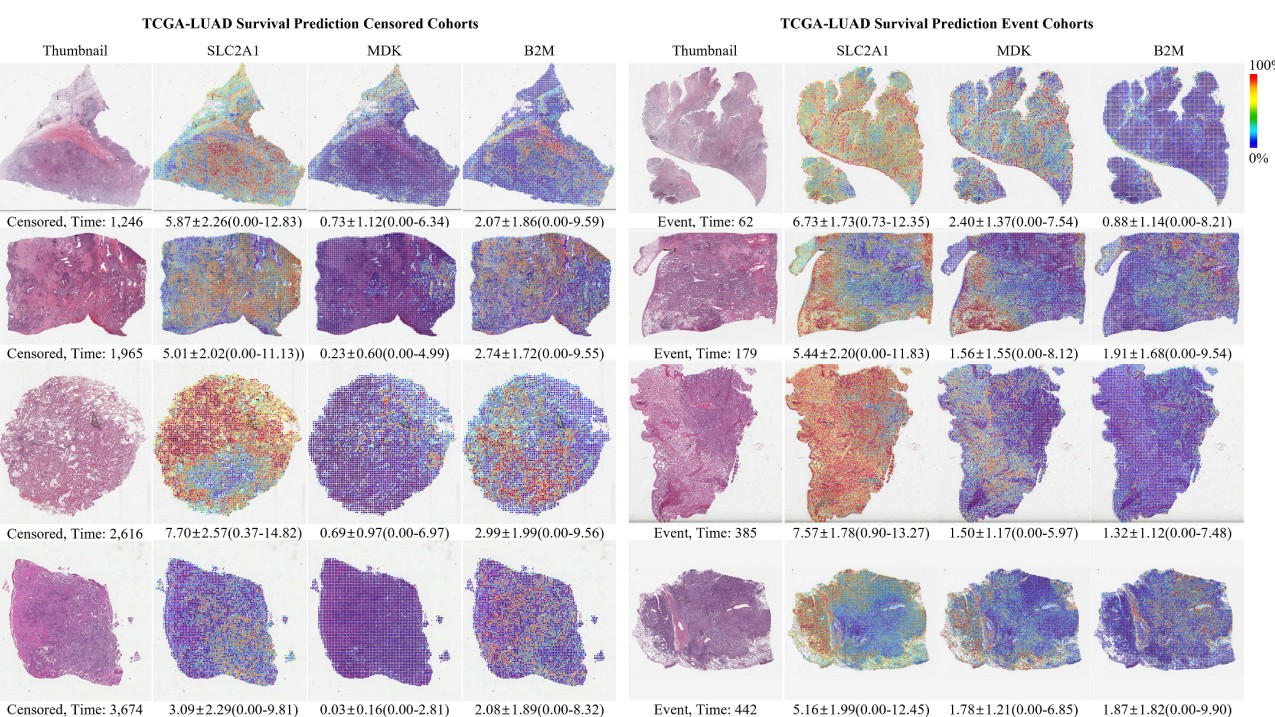

*Figure 8.* Qualitative examples of transcriptomics-guided survival prediction on TCGA-LUAD. For each patient, we show a representative slide thumbnail annotated with survival outcome (event vs. censored) and follow-up time, together with predicted spatial gene expression heatmaps for *SLC2A1*, *MDK*, and *B2M*. The values reported below each heatmap indicate the mean $\pm$ standard deviation with the minimum–maximum range of predicted expression computed across all spots within the slide.

