# OpenReview forum: "HEXST: Hexagonal Shifted-Window Transformer for Spatial Transcriptomics Gene Expression Prediction"
_ICML.cc/2026/Conference — ICML 2026 regular_

### Official Review · Reviewer_cSgb · 2026-03-11

**Soundness:** 3
**Presentation:** 3
**Significance:** 3
**Originality:** 3
**Overall Recommendation:** 4
**Confidence:** 2

**Summary:**

This paper studies spatial gene expression prediction from H&E histology and proposes HEXST, a geometry-aligned Transformer tailored to the approximately hexagonal spot layout of Visium-like ST platforms. Overall, this research’s main contribution comprises a hexagonal shifted-window attention design, a hexagonal rotary positional encoding (HexRoPE), and two training-time components—transcriptomic feature alignment with a pretrained single-cell foundation model and a deviation-matching objective intended to reduce over-smoothing and preserve gene-wise spatial heterogeneity. The paper evaluates the method on seven SpaRED datasets against a reasonably strong set of recent baselines, reports consistent improvements across multiple metrics, and further explores transfer to downstream TCGA tasks for Gleason grading and survival prediction.

**Compliance With Llm Reviewing Policy:**

Affirmed.

**Key Questions For Authors:**

None.

**Limitations:**

The manuscript would benefit from acknowledging that: (i) the main empirical evidence is concentrated on Visium-style hexagonal data and seven preprocessed SpaRED datasets with fixed 128-gene panels; (ii) some proposed components show mixed effects in ablation rather than uniformly improving all metrics; and (iii) the downstream clinical experiments show only modest improvements over the strongest baseline and remain below the bulk-RNA reference in survival prediction. A clearer discussion of these points would make the paper more balanced and would better define the practical scope of the method.

**Strengths And Weaknesses:**

**Strength**: The paper is well motivated and addresses a real modeling mismatch in current ST prediction pipelines: many prior methods use Cartesian or geometry-agnostic locality even though the dominant spot-array platforms are approximately hexagonal. The geometry-aware formulation, including hexagonal windows and HexRoPE, is a coherent and intuitively appealing inductive bias.
**Weakness**: The evidence for broader real-world utility is still somewhat limited. The main benchmark uses seven Visium-based SpaRED datasets with fixed released preprocessing and a dataset-specific target panel of only 128 genes, including spatial smoothing and batch correction performed upstream. This makes the evaluation standardized, but it also narrows the demonstrated scope. The downstream TCGA gains over PEKA are relatively small (e.g., 48.55 vs 48.40 accuracy for PRAD and 0.5970 vs 0.5904 C-index for LUAD), and both image+transcriptomics variants remain below the bulk-RNA upper bound in survival prediction.

---

> ### Author Rebuttal · Authors · 2026-03-31
>
> We thank the reviewer for this valuable feedback. We will clarify the empirical scope and downstream performance interpretations in the manuscript.
>
> ---
>
> __The main empirical evidence is concentrated on Visium-style hexagonal data and seven preprocessed SpaRED datasets with fixed 128-gene panels__
>
> Although our experiments were conducted on the Visium-style SpaRED benchmark with 128 target genes, this setting is more heterogeneous than it may initially appear, and HEXST is not tided to a specific platform.
>
> For each dataset, the 128 genes were independently selected from ~20k genes, with minimal overlaps among datasets (Table R3).
> This indicates that HEXST was not repeatedly evaluated on one shared gene set, but on multiple distinct transcriptomic subspaces.
> In fact, the 128-gene SpaRED setup is a standardized benchmark choice for controlled comparison, not a methodological limitation, as most existing works adopt similar benchmark design [1,2].
>
> Furthermore, as detailed in our response on hexagonal grid generalizability to __vo2L__, HEXST is not coupled to Visium's layout. The slot-based packing scheme operates on raw 2D Cartesian coordinates and accommodates irregular spacing and missing spots without modifications. It was successfully applied to TCGA WSIs organized on a Cartesian grid.
> Additionally, HEXST is inherently scalable to any target gene set size.
>
> * __Table R3: Pairwise overlap of dataset-specific gene panels__
> | |AHSCC|EHPCP1|MMBO|MMBP1|MMBP2|VMB|VLO|
> |-|-|-|-|-|-|-|-|
> |AHSCC|128|17|6|3|4|5|14|
> |EHPCP1|17|128|6|4|5|8|21|
> |MMBO|6|6|128|5|6|4|9|
> |MMBP1|3|4|5|128|64|53|4|
> |MMBP2|4|5|6|64|128|76|10|
> |VMB|5|8|4|53|76|128|13|
> |VLO|14|21|9|4|10|13|128|
>
> ---
>
> __Mixed effects in ablation rather than uniformly improving all metrics__
>
> We agree that the mixed metric improvements warrant discussion. Rather than indicating instability, this reflects the inherent differences between evaluation metrics: $PCC$ reflects linear correlation, $MI$ captures statistical dependence, and $AUC$ focuses on separability.
> As metrics are complementary, improving one does not guarantee proportional gains in others.
>
> Removing $L_{TFA}$ or $L_{DEV}$ relaxes representational constraints, which may allow the model to emphasize broader distributional dependencies ($MI$) or coarse separability ($AUC_{0vNZ}$). However, this consistently degrades $PCC_F$, $PCC_S$, and $AUC_{Q50}$, reducing linear predictive fidelity and spatial consistency.
> Meanwhile, $PCC$ is the most widely adopted primary metric in gene expression prediction, both in machine-learning and in broader biomedical studies.
> Therefore, the full model achieves the most consistent improvements on the primary metrics while maintaining competitive performance across all complementary evaluation criteria.
>
> Nevertheless, we acknowledge that this mixed effect behavior is a limitation of our study, and we will discuss this in the final manuscript.
>
> ---
>
> __The downstream clinical experiments show only modest improvements over the strongest baseline and remain below the bulk-RNA reference in survival prediction__
>
> In PRAD Gleason grading, while accuracy gains appear modest, HEXST substantially improves Cohen’s kappa. Considering the class imbalance (Table 7), kappa better reflects the model's true classification reliability.
>
> Regarding LUAD overall survival prediction, our method underperforms bulk RNA. However, bulk RNA represents the theoretical upper bound, as survival heavily depends on genotypic factors beyond tissue morphology [4,5]. Consequently, it is expected that image-based models will not surpass direct RNA-seq measurements.
>
> Nevertheless, integrating H\&E with HEXST-predicted transcriptomic features achieves prognostic performance approaching RNA-seq analysis. This proves HEXST's practical value as an efficient transcriptomic surrogate in real-world clinical workflows where actual RNA-seq data is not routinely available.
>
> ---
>
> __References__
>
> [1] He, Bryan, et al. "Integrating spatial gene expression and breast tumour morphology via deep learning." Nature biomedical engineering 4.8 (2020): 827-834.
>
> [2] Pan, Shi, Jianan Chen, and Maria Secrier. "Teaching pathology foundation models to accurately predict gene expression with parameter efficient knowledge transfer." International Conference on Medical Image Computing and Computer-Assisted Intervention. Cham: Springer Nature Switzerland, 2025.
>
> [3] Obayashi, Takeshi, and Kengo Kinoshita. "Rank of correlation coefficient as a comparable measure for biological significance of gene coexpression." DNA research 16.5 (2009): 249-260.
>
> [4] Cao, Shaolong, et al. "Estimation of tumor cell total mRNA expression in 15 cancer types predicts disease progression." Nature biotechnology 40.11 (2022): 1624-1633.
>
> [5] Cancer Genome Atlas Research Network. "Integrated genomic analyses of ovarian carcinoma." Nature 474.7353 (2011): 609.

---

> > ### Author Rebuttal · Reviewer_cSgb · 2026-04-05
> >
> > all complete

---

> > > ### Author Response · Authors · 2026-04-06
> > >
> > > We thank the reviewer for acknowledging our response.
> > > We will incorporate this discussion into the final version.

---

### Official Review · Reviewer_vo2L · 2026-03-12

**Soundness:** 2
**Presentation:** 3
**Significance:** 2
**Originality:** 2
**Overall Recommendation:** 3
**Confidence:** 4

**Summary:**

The paper presents HEXST, a Transformer-based architecture designed to predict spatial gene expression directly from H&E histology images. The core innovation lies in adapting the shifted-window attention mechanism and rotary positional embeddings (HexROPE) to specifically align with the hexagonal grid layout typical of spot-based spatial transcriptomics platforms, such as 10x Visium. To prevent the common issue of over-smoothing in spatial predictions, the authors incorporate a deviation-matching training objective and leverage transcriptomic priors from a pre-trained single-cell foundation model. The model is evaluated on seven datasets from the SpaRED benchmark and demonstrates superior performance compared to existing methods, while also showing utility in downstream clinical tasks like survival prediction and tumor grading.

**Compliance With Llm Reviewing Policy:**

Affirmed.

**Final Justification:**

Although the authors have addressed several of my concerns, important issues remain. The rebuttal clarifies that the model can, in principle, be extended beyond hexagonal grids and exhibits some robustness to spatial distortions. However, these discussions primarily focus on slide-level spatial layouts. More critically, the central challenge in modern spatial transcriptomics is to infer gene expression at cellular resolution, where spatial coordinates are continuous and do not follow any predefined lattice. In such settings, the reliance on a hexagonal partitioning scheme—even as a flexible approximation—introduces a geometric inductive bias that is unlikely to be appropriate. As a result, it remains unclear how the proposed framework can be effectively applied to true single-cell or subcellular-resolution data, which limits its broader applicability. Therefore, I maintain my original assessment.

**Key Questions For Authors:**

Generalizability beyond Visium: How can the HEXST architecture be adapted to handle the increasingly common continuous-coordinate, single-cell resolution spatial transcriptomics platforms (e.g., MERFISH [1], Stereo-seq [2]) that inherently do not conform to a hexagonal lattice?

• Robustness to physical distortion: The paper notes the estimation of a representative spacing scale ($s_{spot}$) because real-world grids are imperfect. How sensitive is the HexMSA mechanism to severe physical distortions, tissue tearing, or folding where the rigid hexagonal prior completely breaks down?

• Ablation of geometric vs. semantic priors: The ablation study shows that removing the transcriptomic feature alignment ($\mathcal{L}_{TFA}$) and deviation-matching ($\mathcal{L}_{DEV}$) losses causes performance drops. Have you tested a baseline using standard Cartesian Swin-Transformer attention but retaining these two novel semantic loss functions to isolate exactly how much of the performance gain is purely due to the hexagonal inductive bias versus the semantic losses?

**Limitations:**

Authors should discuss the generalizability of the model beyond the hexagonal grid layout.

**Strengths And Weaknesses:**

Soundness

•	Strengths: The architectural adaptations, specifically HexMSA and HexROPE, are mathematically well-formulated for hexagonal coordinate systems. Furthermore, the empirical validation is comprehensive, utilizing multiple datasets and relevant downstream clinical tasks to demonstrate immediate performance gains.

•	Weaknesses: The core premise is deeply flawed regarding future-proofing and generalizability. The entire architectural design, including window partitioning and slot packing, is strictly hardcoded to the physical hexagonal grid prior of a specific commercial platform (Visium). As the field rapidly advances towards single-cell resolution or continuous-coordinate spatial transcriptomics, these rigid hexagonal constraints render the model inflexible and severely limit its long-term academic and practical value.

Presentation

•	Strengths: The paper is well-written, offering clear mathematical definitions for the coordinate transformations and objective functions. The qualitative heatmaps also effectively illustrate the model's ability to preserve spatial contrast.

•	Weaknesses: The manuscript largely ignores the limitations of tying a deep learning architecture so closely to a proprietary hardware layout. It fails to adequately discuss how the framework would adapt to non-hexagonal or continuous spatial data formats, which is a glaring omission for a modern computational biology paper.

Significance

•	Strengths: Predicting spatial expression from ubiquitous H&E slides remains a highly valuable goal for computational pathology and clinical deployment.

•	Weaknesses: The significance is fundamentally bottlenecked by the model's over-specialization. By over-fitting the inductive bias to Visium's specific hexagonal lattice, the model's impact is sharply restricted to retrospective analyses of older, platform-specific datasets.

Originality

•	Strengths: Extending Rotary Positional Encoding (ROPE) to a hexagonal coordinate system (HexROPE) is a clever technical adaptation.

•	Weaknesses: The originality is primarily incremental engineering. It forces existing Swin-Transformer concepts onto a hexagonal grid. While combining a deviation-matching loss with single-cell foundation model priors is effective, it does not compensate for the fatal flaw in geometric generalizability.

---

> ### Author Rebuttal · Authors · 2026-03-31
>
> We thank the reviewer for this valuable feedback. We would like to address the reviewer’s comments as follows. We will incorporate the discussion into the final manuscript.
>
> ---
>
> __Generalizability beyond Visium__
>
> We believe that the concern raised by the reviewer regarding the generalizability of the hexagonal grid prior is a very important point, and we would like to clarify this more explicitly.
> While it is also a practical reason for adopting the hexagonal structure that a large portion of publicly available spatial transcriptomics data, such as 10x Visium, utilizes hexagonal grids, the core objective of this study is not to design a model that depends on the hexagonal grid of a specific platform. Rather, our objective is to enable more effective spatial context modeling.
>
> Among regular tessellations, hexagonal partition provides more uniform angular coverage through its 6-directional symmetry. This geometric property is universally applicable and is not limited to a specific coordinate system or data format.
> To avoid hard coupling to a specific platform, we designed the hexagonal partition using a slot-based packing scheme (as described in Appendix A). In this scheme, a fixed number of slots is defined within each window, and spots located inside the window are assigned to those predefined slots based on geometric distance. This scheme is not restricted to hexagonal grids and can naturally accommodate irregular inter-spot distances and missing spots within a window.
>
> For higher-resolution platforms, the spot- or pixel-level layout is typically represented in Cartesian coordinates. As the spatial layout remains regular, the same slot-based packing scheme can be used to construct hexagonal window in a highly consistent manner.
> Notably, we empirically validated this cross-grid generalizability in our downstream tasks using the TCGA datasets. WSI data in the TCGA datasets follow a Cartesian grid, not a hexagonal grid. By utilizing the proposed slot-based packing scheme, the model trained solely on hexagonal grid-based data was directly applied to TCGA data without requiring architectural modifications. The results confirm the broad applicability and generalizability of the proposed hexagonal window with slot-based packing scheme.
>
> From this perspective, the hexagonal structure in HEXST should not be interpreted as a platform-specific lattice assumption, but as a spatial partitioning strategy.
> In the current implementation, scaling was applied based on Visium's inter-spot distances, this is merely an implementation choice rather than a fundamental limitation.
> Adjusting coordinate normalization allows defining the same hexagonal windows naturally in continuous coordinate systems.
> For example, the framework can be extended by partitioning the spatial space into hexagonal tiling while preserving continuous coordinates and assigning each point to its corresponding window, without requiring major modifications to the model architecture.
> Therefore, this framework can be directly extended to imaging-based spatial transcriptomics platforms with continuous coordinates such as MERFISH or Stereo-seq. It would also be possible to maintain the same hexagonal partition while replacing the backbone with an alternative architecture such as graph neural networks.
> We fully agree that the discussion on generalizability raised by the reviewer is important, and we will make this discussion clearer in the final manuscript.
>
> ---
>
> __Robustness against physical distortions__
>
> We would like to note that cases where spot coordinates are misaligned or spots are missing can be resolved using the slot-packing approach.
> However, if physical distortions such as tissue folding or tearing occur during the sample preparation process, the observed coordinates fail to accurately reflect the true biological spatial relationships, which can be a fundamental challenge for spatially-aware models.
>
> Nevertheless, because our model operates based on relative spatial relationships within each window rather than assuming a strict global grid structure, its sensitivity to moderate levels of geometric misalignment is mitigated, as long as the underlying spatial relationships are not completely disrupted. Therefore, while it cannot completely correct for severe topological distortions, it can operate relatively stably even in the presence of spatial structural irregularities, and it functions reliably even on non-hexagonal grids.
>
> ---
>
> __Ablation of geometric vs. semantic priors__
>
> To disentangle geometric inductive biases from semantic losses, we compared our hexagonal window + HexRoPE against (i) hexagonal window + 2D Cartesian RoPE and (ii) square window + 2D Cartesian RoPE.
> The experimental results confirm that both the hexagonal window and HexRoPE drive consistent performance improvements across all evaluation metrics.
>
> To avoid redundancy, we refer the reviewer to our response to Reviewer __dhot__ for detailed results and analysis.

---

> > ### Author Rebuttal · Reviewer_vo2L · 2026-04-02
> >
> > Although the authors have addressed several of my concerns, important issues remain. The rebuttal clarifies that the model can, in principle, be extended beyond hexagonal grids and exhibits some robustness to spatial distortions. However, these discussions primarily focus on slide-level spatial layouts. More critically, the central challenge in modern spatial transcriptomics is to infer gene expression at cellular resolution, where spatial coordinates are continuous and do not follow any predefined lattice. In such settings, the reliance on a hexagonal partitioning scheme—even as a flexible approximation—introduces a geometric inductive bias that is unlikely to be appropriate. As a result, it remains unclear how the proposed framework can be effectively applied to true single-cell or subcellular-resolution data, which limits its broader applicability. Therefore, I maintain my original assessment.

---

> > > ### Author Response · Authors · 2026-04-06
> > >
> > > We sincerely thank the reviewer for this constructive feedback.
> > > We agree with the reviewer that handling continuous, lattice-free spatial coordinates at cellular or subcellular resolution is a critical challenge in modern spatial transcriptomics.
> > >
> > > However, we respectfully disagree with the reviewer that our hexagonal partitioning introduces an overly restrictive or inappropriate prior. In continuous-coordinate spatial transcriptomics, processing millions of unconstrained cells simultaneously is computationally and memory intensive. Consequently, most existing methods impose some form of artificial spatial partitioning or local neighborhood grouping to make data tractable.
> > > For example, CellTransformer [1] constructs local neighborhoods by considering surrounding cells within a square window centered on each reference cell.
> > > SToFM [2] divides the entire slide into multiple sub-slices and performs learning over these local cell sets.
> > > SpatialEx [3] groups cells using a hypergraph structure to capture local relationships.
> > > Additionally, Yarlagadda et al. [4] employ a 2D convolutional architecture, which inherently induces a Cartesian grid-based square partition over intermediate feature maps.
> > > When examining these partitioning strategies, it is clear that square-based partitioning schemes are widely utilized not because continuous cellular data inherently follows a square lattice, but strictly as a necessary computational design choice to effectively utilize spatial information in the data.
> > >
> > > We would like to emphasize that our hexagonal partitioning and positional encoding should be interpreted as one such design choice.
> > > The hexagonal structure offers more uniform directional connectivity, which can help represent balanced local information in continuous space.
> > > In continuous coordinate settings, the slot packing within each hex window and the HexRoPE can be naturally extended from discrete formulations to continuous spatial environments.
> > > Therefore, we believe it is more appropriate to interpret our approach as a practical means of utilizing spatial information, rather than as a prior or constraint imposed on the data.
> > >
> > > We believe that providing a clearer description of this extension to continuous settings would further strengthen the manuscript, and we will elaborate on this more explicitly in the final revised version.
> > >
> > > ---
> > >
> > > __References__
> > >
> > > [1] Lee, Alex J., et al. "Data-driven fine-grained region discovery in the mouse brain with transformers." Nature Communications 16.1 (2025): 8536.
> > >
> > > [2] Zhao, Suyuan, et al. "Stofm: a multi-scale foundation model for spatial transcriptomics." arXiv preprint arXiv:2507.11588 (2025).
> > >
> > > [3] Liu, Yonghao, et al. "High-parameter spatial multi-omics through histology-anchored integration." Nature Methods (2025): 1-14.
> > >
> > > [4] Yarlagadda, Dig Vijay Kumar, Joan Massagué, and Christina Leslie. "Discrete representation learning for modeling imaging-based spatial transcriptomics data." Proceedings of the IEEE/CVF International Conference on Computer Vision. 2023.

---

### Official Review · Reviewer_dhot · 2026-03-13

**Soundness:** 3
**Presentation:** 3
**Significance:** 3
**Originality:** 3
**Overall Recommendation:** 5
**Confidence:** 4

**Summary:**

This paper proposes HEXST, a Transformer architecture that aligns with the hexagonal geometry of Visium spatial transcriptomics. The key contributions are: hexagonal shifted-window attention (HexMSA), hexagonal rotary positional encoding (HexRoPE) and transcriptomic feature alignment using single cell foudantion model embeddings. The method is evaluated on seven SpaRED benchmark datasets and two TCGA downstream tasks.

**Compliance With Llm Reviewing Policy:**

Affirmed.

**Final Justification:**

Thanks authors for the rebuttal. My concerns on cross species transfer and extensibility to new technology have been addressed. I have  updated my scores accordingly

**Key Questions For Authors:**

- See my weakness
- Additionally, since higher resolution spatial assays are more widely used today (such as visiumHD, xenium, cosmx), is the method adaptable to those assays?

**Limitations:**

the authors adequately discussed the limitations

**Strengths And Weaknesses:**

- Strength

  - The core motivation is clear and well articulated. The paper provides a clean formalization of hexagonal window partitioning, slot-based packing, and shifting.

  - The method is reasonably well engineered. In particular, the use of rotary positional encoding adapted to hexagonal cube coordinates is a thoughtful design choice.
  - The empirical results are promising overall, with consistent improvements across multiple metrics and datasets.
- Weakness

  - My main critique is about ablation experiments, While I appreciate that comprehensive loss term ablations are provided (eg. Table 2), several important architectural ablations are missing, including
    - hexagonal windows vs square windows

    - HexRoPE ablation with standard positional encoding

    - etc
  - It appears that the majority of datasets are from mouse. scFoundation used for transcriptomic feature alignment was trained exclusively on human data. The paper does not discuss how well this cross species transfer works or whether the alignment signal is meaningful for the mouse datasets.

---

> ### Author Rebuttal · Authors · 2026-03-31
>
> We thank the reviewer for this valuable feedback and will incorporate the discussion into the final manuscript.
>
> ---
>
> __Architectural ablations__
>
> As pointed out, the current manuscript lacks architectural ablations compared to the loss-focused ablation.
> To address this, we compared our proposed hexagonal window + HexRoPE against
> (i) hexagonal window + 2D Cartesian RoPE and
> (ii) square window + 2D Cartesian RoPE
> while keeping other settings unchanged.
> 2D Cartesian RoPE was implemented using relative (x, y) positions from the window center in the Cartesian coordinate system, followed by applying 2D RoPE based on these relative positions.
>
> Table R1 shows that both the hexagonal window and HexRoPE lead to consistent improvements across all metrics. For example, replacing the square window with the hexagonal window improves $PCC_F$ from 0.3690 to 0.3993, and introducing HexRoPE provides an additional gain of +0.0234 in $PCC_F$.
>
> These results confirm that the hexagonal window is essential for spatial neighborhood modeling, and HexRoPE provides additional gains through geometrically consistent positional information. This ablation study will be added to the final manuscript.
>
> * __Table R1: Architectural Ablation study of HEXST__
> |Window|Positional Encoding|$PCC_F$↑|$PCC_S$↑|$MI_F$↑|$AUC_{0vNZ}$↑|$AUC_{Q50}$↑|
> |-|-|-|-|-|-|-|
> |Square Window|2D Cartesian RoPE|0.3690±0.15|0.7421±0.09|0.1405±0.04|0.6617±0.04|0.6941±0.05|
> |Hexagonal Window|2D Cartesian RoPE|0.3993±0.11|0.7508±0.09|0.1521±0.06|0.6670±0.03|0.7156±0.03|
> |Hexagonal Window|HexRoPE|0.4227±0.12|0.7697±0.08|0.1540±0.05|0.6820±0.05|0.7227±0.03|
>
> ---
>
> __Cross-species transcriptomic feature alignment using scFoundation__
>
> As the reviewer noted, scFoundation was originally trained on human genomic data while our experiments include mouse datasets.
> We understand that this cross-species difference may raise concerns.
> However, it is widely recognized that orthologous genes are conserved between humans and mice at the sequence, expression, and functional levels [1]. Therefore, cross-species transcriptomic alignment remains biologically valid.
>
> Examining the overlap between each dataset-specific gene panel and the scFoundation vocabulary (Table R2), we found that 118-122 genes were matched across all datasets. Given this substantial overlap of orthologous genes and conserved gene expression patterns between humans and mice [1], scFoundation embeddings remain informative and provide sufficient signals for training HEXST, as supported by the consistent performance gains on the mouse datasets.
> Unmatched genes are mostly species-specific or non-coding RNAs (e.g., MALAT1, pseudogenes) excluded by scFoundation.
>
> Importantly, while scFoundation is used for feature extraction, HEXST predicts all 128 target genes. Since transcriptomic features use distributed representations, information from excluded genes can be indirectly captured, imposing no practical limitation.
>
> We acknowledge that the impact of excluded genes was not explicitly quantified and leave the effect of vocabulary size to future work.
>
> * __Table R2: Overlap between dataset-specific 128-gene panels and the scFoundation vocabulary (19,264 genes).__
> |Dataset|Matched|Missing|Missing genes|
> |-|-|-|-|
> |AHSCC|122|6|AES,H2AFV,IGHG1,IGKC,LOR,MALAT1|
> |EHPCP1|118|10|C19ORF48,C1ORF21,H2AFJ,H2AFZ,MALAT1,PART1,SARS,SNHG19,SNHG25,SNHG8|
> |MMBO|120|8|2310022B05RIK,CAR3,COX8B,HBA-A2,HBB-BS,HIST1H1A,MT1,SEPT5|
> |MMBP1|121|7|1110008P14RIK,2010300C02RIK,CAR2,SCD2,SEPT4,SEPT5,TRF|
> |MMBP2|120|8|1110008P14RIK,CAR2,CDR1OS,MT1,QK,SCD1,SCD2,TRF|
> |VMB|118|10|1110008P14RIK,CAR2,MALAT1,MEG3,OIP5OS1,PNMAL2,QK,SCD2,SEPT4,TRF|
> |VLO|122|6|H1F0,MALAT1,MIAT,NORAD,SEPT2,SNHG25|
>
> ---
>
> __Adaptability to High-Resolution Spatial Assays__
>
> We agree with the reviewer that higher-resolution platforms, such as Visium HD, Xenium, and CosMx, are becoming increasingly widespread and their coordinate systems differ from the Visium platform on which HEXST was originally developed.
> However, HEXST, in particular the hexagonal window and HexRoPE, is not restricted to a specific platform and is designed to be adaptable to higher-resolution platforms for the following reasons.
> First, the hexagonal window partition is not bound to a pre-existing hardware grid. Through spatial coordinate normalization, any continuous coordinate space can be partitioned using hexagonal tiling.
> Second, the slot-based packing does not rely on a predefined grid and can accommodate non-uniform spot densities or missing regions.
> Third, downstream tasks showed that models trained on hexagonal grid-based data can be directly applied to Cartesian grid-based TCGA WSI data without architectural modifications.
>
> To avoid redundancy, we refer the reviewer to our response to Reviewer __vo2L__ for more detailed discussion.
>
> ---
>
> __References__
>
> [1] Zheng-Bradley, Xiangqun, et al. "Large scale comparison of global gene expression patterns in human and mouse." Genome biology 11.12 (2010): R124.

---

> > ### Author Rebuttal · Reviewer_dhot · 2026-04-04
> >
> > The authors have fully addressed my concerns. That said, I still think including a baseline comparison against cross-species scFMs would strengthen the claims (such as universal cell embedding [1] or nicheformer [2] ), though this could be left as a future extension.
> >
> > [1] https://www.biorxiv.org/content/10.1101/2023.11.28.568918v1
> > [2] https://www.nature.com/articles/s41592-025-02814-z

---

> > > ### Author Response · Authors · 2026-04-06
> > >
> > > We agree with the reviewer that comparing with recent cross-species foundation models is a meaningful extension.
> > > In particular, such comparisons can be incorporated into our framework without major architectural modifications by applying appropriate gene mapping according to each model’s gene vocabulary.
> > > In future work, we plan to extend our approach to a wider range of genes, species, platforms, and omics modalities.
> > > We believe that it would be beneficial to analyze how performance varies depending on the pretraining data of each foundation model.
> > >
> > > We thank the reviewer for the constructive feedback.
> > > We will incorporate this discussion into the final version.

---

### Official Review · Reviewer_tFbn · 2026-03-13

**Soundness:** 4
**Presentation:** 4
**Significance:** 3
**Originality:** 4
**Overall Recommendation:** 5
**Confidence:** 3

**Summary:**

HEXST predicts spatial gene expression from histograms. Unlike previous models which assumed cartesian or geomerty-agnostic locality, HEXST uses a geometric-aligned transformer to avoid over-smoothed gene expression caused by point-wise regression inconsistant with the sot-array platform hexagonal sampling. The main improvement is to add contrastive-sensitive differential objective and transcriptomic priors during training.

**Compliance With Llm Reviewing Policy:**

Affirmed.

**Final Justification:**

Overall, I recommend a 5 (Accept) for this paper. The authors tackle a highly specific and practical challenge in computational biology: adapting transformer architectures to match the non-Cartesian, hexagonal sampling geometry of widespread spatial transcriptomics platforms like Visium.

The technical contributions are sound. The introduction of HexMSA and HexROPE provides an original, mathematically rigorous architectural inductive bias that elegantly resolves the anisotropic receptive field issues seen in standard Cartesian vision models. Moreover, the novel deviation-matching loss function successfully addresses the over-smoothing from point-wise regression, a known failure in the field, by explicitly penalizing the loss of high-frequency spatial contrast. These methodological innovations result in consistent empirical excellence across 7 distinct datasets and multiple evaluation metrics.

The main limitation lies in the paper's biological framing. While the model mathematically preserves "gene-specific spatial heterogeneity" better than baselines, the downstream biological utility of recovering this is not as obvious. However, this is a gap in interpretation rather than a fundamental flaw in the model's design or evaluation.

Given the clear algorithmic advancements, the strong empirical validation, and the high relevance of the problem, the technical merits of the paper support acceptance. If the authors can quantify the biological value of their work, this paper will be an excellent addition to the conference.

**Key Questions For Authors:**

Could the author expand on the biological importance of not having a overly smoothed gene profile using the point-wise regression as the previous works?

**Limitations:**

yes

**Strengths And Weaknesses:**

### Strengths
* Clear & Strong Architectures: The formulation of the HexMSA and HexROPE is original and mathematically rigorous. By mapping non-Cartesian spot coordinates to a 3-axis cube coordinate system and using Voronoi-style partitioning, the model perfectly aligns its field with the actual physical sampling geometry of spot-array platforms like Visium. This directly resolves the anisotropic receptive field issues of Cartesian-based vision models.
* Novel Objective Function: The paper tackles a well-known failure mode in spatial gene expression prediction: over-smoothing caused by standard point-wise regression (MSE). The proposed deviation-matching loss is a clever regularization technique that computes and standardizes deviations in the embedding space to preserve high-frequency spatial contrast.
* Effective Use of Foundation Models: The transcriptomic feature alignment injects biological priors by aligning intermediate representations with a pretrained single-cell foundation model (scFoundation) during training only, avoiding inference-time overhead.
* Comprehensive Evaluation: The benchmarking is exceptionally thorough. As shown in Table 1, HEXST uniformly improves upon a strong baseline suite across 7 SpaRED datasets and 5 distinct metrics. The ablation study (Table 2) effectively isolates the contribution of each proposed loss term, and the mathematical formulations are rigorous and concise.

### Weaknesses
* Unclear Downstream Biological Utility of Preserved Heterogeneity: The paper’s core biological claim is that it preserves "gene-specific spatial heterogeneity" better than previous methods, avoiding over-smoothed gene profiles. While the qualitative heatmaps (Figures 4 and 7) and metrics support this, the actual biological utility of recovering this fine-grained variance is not deeply explored. A biologist using this tool would want to know if this sharper contrast leads to better downstream biological discoveries.

---

> ### Author Rebuttal · Authors · 2026-03-31
>
> We thank the reviewer for acknowledging the algorithmic contributions of this study. We would like to address the core questions as follows.
>
> ---
>
> __Expand on the biological importance of not having a overly smoothed gene profile__
>
> Important biological signals in spatial transcriptomics data lie not only in the absolute expression values but also in spatial variations such as spatial gradients, boundaries, and contrast [1,2,3].
> For example, tumor-stroma boundaries, immune niches, and other histological structures often manifest as sharp, abrupt changes in gene expression rather than gradual transitions. Accurate delineation of these spatial features is essential for identifying and analyzing localized cell-to-cell and microenvironmental interactions, and for discovering spatially restricted cellular states and biomarkers.
>
> However, existing point-wise regression-based approaches are typically trained to minimize per-spot error, which naturally encourages predictions toward the local mean.
> This tendency arises from the sparse and noisy nature of gene expression data, combined with the inherent mechanism of models that aggregate neighboring spots.
> From the perspective of actual biological interpretation, this phenomenon can cause problems where important structural boundaries, such as the tumor–stroma boundary, become blurred, making it difficult to distinguish pathologically important compartments.
>
> The deviation-matching objective proposed in this study was designed to mitigate this limitation. By learning the relative deviation between spots rather than simply fitting absolute expression values, it prevents the model from collapsing into local averaging and helps preserve spatial contrast.
> Our qualitative results confirm that HEXST produces sharper boundaries and better preserves gene-specific spatial patterns compared to existing methods. We believe this is a significant improvement as it maintains biologically meaningful structures.
>
> We appreciate the reviewer for this constructive feedback. We will incorporate this extended explanation in the final manuscript.
>
> ---
>
> __References__
>
> [1] Feng, Yu, et al. "Spatially organized tumor-stroma boundary determines the efficacy of immunotherapy in colorectal cancer patients." Nature communications 15.1 (2024): 10259.
>
> [2] Takano, Yuma, et al. "Spatially resolved gene expression profiling of tumor microenvironment reveals key steps of lung adenocarcinoma development." Nature Communications 15.1 (2024): 10637.
>
> [3] Kueckelhaus, Jan, et al. "Inferring histology-associated gene expression gradients in spatial transcriptomic studies." Nature Communications 15.1 (2024): 7280.

---

> > ### Author Rebuttal · Reviewer_tFbn · 2026-04-04
> >
> > I appreciate the detailed explanation on the biological significance of not having a overly smoothed gene profile, and their results supported their claim quantitatively and qualitatively. Thus I will keep my score in the positive.

---

> > > ### Author Response · Authors · 2026-04-06
> > >
> > > We thank the reviewer for acknowledging our response.
> > > We will incorporate this discussion into the final version.

---

### Decision · Program_Chairs · 2026-04-30

**Decision:**

Accept (regular)

**Comment:**

The authors tackle a highly specific and practical challenge in computational biology: adapting transformer architectures to match the non-Cartesian, hexagonal sampling geometry of widespread spatial transcriptomics platforms like Visium. reviewers were for the most part happy with the rebuttal and recognized the technical contributions of the paper in terms of architecture